# Keep your distance: learning dispersed embeddings on $\mathbb{S}_m$

**Evgeniia Tokarchuk** *evgeniia@tokarch.uk*
*Language Technology Lab*
*University of Amsterdam*

**Hua Chang Bakker** *hua.chang.bakker@student.uva.nl*
*University of Amsterdam*

**Vlad Niculae** *v.niculae@uva.nl*
*Language Technology Lab*
*University of Amsterdam*

**Reviewed on OpenReview:** *https: // openreview. net/ forum? id= 5JIQE6HcTd*

## Abstract

Learning well-separated features in high-dimensional spaces, such as text or image *embeddings*, is crucial for many machine learning applications. Achieving such separation can be effectively accomplished through the *dispersion* of embeddings, where unrelated vectors are pushed apart as much as possible. By constraining features to be on a *hypersphere*, we can connect dispersion to well-studied problems in mathematics and physics, where optimal solutions are known for limited low-dimensional cases. However, in representation learning we typically deal with a large number of features in high-dimensional space, and moreover, dispersion is usually traded off with some other task-oriented training objective, making existing theoretical and numerical solutions inapplicable. Therefore, it is common to rely on gradient-based methods to encourage dispersion, usually by minimizing some function of the pairwise distances. In this work, we first give an overview of existing methods from disconnected literature, making new connections and highlighting similarities. Next, we introduce some new angles. We propose to reinterpret pairwise dispersion using a maximum mean discrepancy (MMD) motivation. We then propose an online variant of the celebrated Lloyd's algorithm, of K-Means fame, as an effective alternative regularizer for dispersion on generic domains. Finally, we revise and empirically assess sliced regularizers that directly exploit properties of the hypersphere, proposing a new, simple but effective one. Our experiments show the importance of dispersion in image classification and natural language processing tasks, and how algorithms exhibit different trade-offs in different regimes.

## 1 Introduction

Dispersion, sometimes referred to as spreading or uniformity, is the property that a collection of points in a domain cover the domain well, with no two points being too close or too far. When the domain is a hypersphere, a useful space for representing embeddings of text (Meng et al., 2019) and images (Liu et al., 2017; Mettes et al., 2019), optimal dispersion is challenging. Embedding clumping, *i.e.*, occurrence of semantically distant embeddings that are close to each other in terms of distance metric, is a known problem, and it has been shown before that it negatively impacts the performance of the downstream tasks, such as image classification (Wang & Isola, 2020; Liu et al., 2021; Trosten et al., 2023), image generation (Liu et al., 2021), text classification (Wang & Isola, 2020) and text generation (Tokarchuk & Niculae, 2024). Beyond data embeddings, neural network layer weights have been shown to also benefit from dispersion (Liu et al., 2018; 2021; Wang et al., 2021).

In general, the problem of dispersing $n$ points on the surface of a $m$-dimensional sphere, such that the angular distance between any two points is maximal, is an open mathematical problem known as the Tammes problem (Tammes, 1930). The optimal solutions for this problem are only known for small values of $m$ and $n$ (Fejes Tóth, 1943; Danzer, 1986; van der Waerden & Schütte, 1951; Robinson, 1961; Musin & Tarasov, 2012; 2015). The Tammes problem can also be formulated as a problem of finding an optimal spherical code (Conway et al., 1999; Cohn, 2024). However, in machine learning applications we typically deal with a large number of dimensions and many points. Moreover, perfect dispersion may not be optimal and may hurt the task-specific learning objective, so in learning applications we prefer being able to trade off between a task loss and a dispersion regularizer. For these reasons, we cannot directly use classical results, and instead focus on hyperspherical dispersion as a gradient-based optimization of a regularization function.

**Contributions.** In this work, we study the regularization approach to dispersion in machine learning applications, proposing new algorithms as well as casting new light on known ones. We survey a number of dispersion objectives used in the literature and reveal relationships between them (§2.4). We then show a new and fundamental connection between kernel-based dispersion and maximum mean discrepancy (MMD, Gretton et al., 2012) with a uniform measure (§3.1). Further, we adapt Lloyd's algorithm (Lloyd, 1982) as a stochastic dispersion regularizer (§3.2), and suggest an effective regularizer based on slicing dispersion along great circles (§3.3). We evaluate (§4) old and new methods on synthetic small and large scale problems, as well as real-world large-scale applications in computer vision and natural language processing, revealing different trade-offs and throughout confirming the importance of representation dispersion for task performance. Additionally, we highlight that using Riemannian optimization (Bonnabel, 2013; Becigneul & Ganea, 2019) on the hypersphere, rather than projecting Euclidean gradient updates, benefits dispersion and overall task performance. A reusable library for spherical dispersion is available as open-source software: https://github.com/ltl-uva/ledoh-torch

## 2 Background: Dispersion on the Hypersphere

### 2.1 Notation

We denote by $\mathbb{S}_m$ the $m-1$-dimensional hypersphere embedded in $\mathbb{R}^m$, *i.e.*, $\mathbb{S}_m = \{x \in \mathbb{R}^m \mid \|x\| = 1\}$. For $u, v \in \mathbb{R}^m$ we denote their Euclidean inner product by $\langle u, v \rangle := \sum_{i=1}^m u_i v_i$. The hypersphere is an embedded Riemannian submanifold of $\mathbb{R}^m$. The tangent space of the hypersphere at a point $x$ is $T_x\mathbb{S}_m := \{v \in \mathbb{R}^m \mid \langle x, v \rangle = 0\}$, and the Riemannian inner product on it is inherited from $\mathbb{R}^m$, *i.e.*, for $u, v \in T_x\mathbb{S}_m, \langle u, v \rangle_x := \langle u, v \rangle$. The geodesic distance or angular distance on $\mathbb{S}_m$ is denoted by $d_{\mathbb{S}}(x, x') := \arccos(\langle x, x' \rangle)$, in contrast with the chordal distance, which is the Euclidean distance in ambient space $d_{\mathbb{R}}(x, x') := \|x - x'\|$. As a special case, for circles ($m = 2$) it is more convenient to work in an isomorphic angular parametrization, *i.e.*, $\mathbb{S}_2$ is isomorphic to $\{\theta \mid -\pi \leq \theta < \pi\} \subset \mathbb{R}$ with $d_{\mathbb{S}}(\theta, \theta') = |\theta - \theta'|$, and the embedding is $\theta \mapsto (\cos\theta, \sin\theta)$. We reserve the use of Greek letters $\tau, \theta, \phi$ for 1-d angles. We denote by $\Pi_n$ the set of permutations of $(1, \ldots, n)$.

We denote as $X = (x_1, \ldots, x_n)$ an (ordered) collection, configuration, or design of $n$ points on the same sphere, *i.e.*, each $x_i \in \mathbb{S}_m$. Sans-serif capitals, *e.g.*, Y, denote random variables.

### 2.2 Riemannian Optimization

Since the configurations we are considering consists of points on a sphere, the natural framework for optimization is Riemannian optimization, taking into account the geometry of the sphere $\mathbb{S}_m$. We briefly describe and illustrate the method for $\mathbb{S}_m$ as an embedded submanifold of $\mathbb{R}^m$. Given a point $x \in \mathbb{S}_m$ and a function $F : \mathbb{S}_m \to \mathbb{R}$, the Riemannian gradient is the projection of the standard (Euclidean) gradient onto the tangent plane at $x$: (Figure 1)

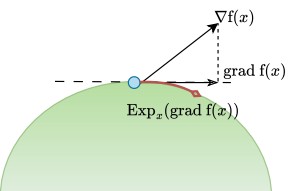

**Figure 1:** Euclidean vs. Riemannian optimization.

$$\operatorname{grad}_x F(x) = \operatorname{proj}_{T_x\mathbb{S}_m}(\nabla_x F(x)) = (I - xx^\top)\nabla_x F(x).$$

Vectors tangent at $x$ can be interpreted as directions of travel for $x$ along the manifold. This intuition gives the Riemannian gradient method:

$$x^{(t+1)} = \mathrm{retr}_{x^{(t)}}(\lambda^{(t)} \mathrm{grad}_x F(x^{(t)})),$$

where $\lambda^{(t)}$ is a sequence of step sizes, and the retraction $\mathrm{retr}_x$ maps tangent vectors back onto the surface. The canonical retraction is the exponential map $\mathrm{retr}_x^{(\mathrm{exp})}(v) = \mathrm{Exp}_x(v) = \cos(\|v\|)x + \sin(\|v\|)\frac{v}{\|v\|}$; it moves from $x$ along the sphere surface with constant velocity $v$ for one unit of time. For optimization, approximate retractions such as the projection-based one also work and can be implemented slightly faster: $\mathrm{retr}_x^{(\mathrm{proj})}(v) = (x+v)/\|x+v\|$. Riemannian gradient methods are due to Luenberger (1972); Lichnewsky (1979); Gabay (1982). Stochastic versions where the gradient is replaced by an estimation are studied by Bonnabel (2013), and accelerated versions like Riemannian Adam are due to Becigneul & Ganea (2019). While some works optimize spherical objectives ignoring the geometry and applying Euclidean gradient descent in $\mathbb{R}^m$ interspersed with projection back onto $\mathbb{S}_m$, we note this is ill-defined: while formally it resembles the projected gradient method, $\mathbb{S}_m$ is a non-convex subset of $\mathbb{R}^m$. Projection is not defined at the origin of $\mathbb{R}^m$, and it exhibits quick jumps around the origin. Despite also being convergent for certain simple objectives (Vu et al., 2019) and being often used, we experimentally show in Appendix E.3 that projected gradient is less well-behaved than Riemannian gradient for our purposes, and therefore stick to the natural choice of Riemannian optimization.

## 2.3 Measuring Dispersion

Intuitively, we say a configuration of $n$ points on $\mathbb{S}_m$ is dispersed if the points are as well spread out, with each point being neither too close nor too far from its neighbors, and therefore covering the surface well. In coding theory, maximally-dispersed configurations form optimal *spherical codes*, and can be used to discretize continuous spherical data. The problem of finding such optimal configurations is also known, equivalently, as the sphere packing problem (Conway et al., 1999) and the Tammes problem (Tammes, 1930), and its general form is hard, with solutions found only for special values of $n$ and $m$. Approximations can be found with continuous optimization methods, which also allow us to tackle multi-task objectives, where we seek to balance dispersion with some other task of interest. In order to develop dispersion regularizers that can be efficiently optimized, we will first review the traditionally used measures to quantify dispersion in the theoretical contexts listed.

**Minimum distance.** The Tammes problem (Tammes, 1930) is the problem of finding a configuration that maximizes the minimum distance between all pairs of points. A natural measure of dispersion would then be:

$$d_{\min}(X) = \min_{1 \le i \ne j \le n} d(x_i, x_j), \tag{1}$$

where $d$ is a distance. In the sequel we will always define $d_{\min}$ in terms of the geodesic distance $d_{\mathbb{S}}$. As a metric, however, $d_{\min}$ may be less useful for comparing dispersion of approximate solutions, as it cannot distinguish between a configuration with $n-1$ well-dispersed points and one $\varepsilon$-close from another, from a configuration where all points are $\varepsilon$-close.

**Spherical variance.** Seeing the points $X$ as vectors in $\mathbb{R}^m$, we may consider their average $\mu = (\sum_{i=1}^n x_i)/n$. This average is a vector lying inside the unit ball, and its length, sometimes called *mean resultant length*, gives a variance estimator studied in directional statistics (Jammalamadaka & Sengupta, 2001; Mardia, 1975):

$$\mathrm{svar}(X) = 1 - \|\mu\|, \text{ where } \mu = \frac{1}{n}\sum_{i=1}^n x_i. \tag{2}$$

If the points are highly concentrated, $\mu$ will be close to the surface of the sphere, and so $\mathrm{svar}(X)$ would be close to zero. When points are dispersed, their directions cancel out, $\mu$ is close to the origin, and so $\mathrm{svar}(X)$ is close to 1. Since $\mu$ must be in the unit ball, $\mathrm{svar}(X)$ must be between 0 and 1.

Unlike $d_{\min}$, configurations maximizing svar are not necessarily well dispersed: the spherical variance of $X = (x, x, -x, -x)$ is 1, but the minimum distance is 0 for any $x$. Nevertheless, its computational efficient and highly global nature make it a useful metric, in ways that can complement minimum distance.

## 2.4 Optimizing for Dispersion

In this section we review some of the methods for numerical optimization of dispersion in previous work. Such methods are almost exclusively pairwise in nature, leading to at least quadratic complexity. Along the way, we make some new connections between the methods.

**Closest-point objectives.** It may be tempting to optimize $d_{\min}(X)$ directly, as this is a definitional measure of dispersion. However, the gradient of $d_{\min}(X)$ is zero for almost all points, with the exception of the closest pair,[1] leading to slow convergence, as pointed out by Mettes et al. (2019) and Liu et al. (2018). An alternative is to consider the closest point to every point in the configuration. The resulting *max-min* (MM) objective appears in several works:

$$\mathcal{L}_{\text{MM}}(X) = -\frac{1}{n} \sum_{i=1}^{n} \left( \min_{j \neq i} d(x_i, x_j) \right). \tag{3}$$

Mettes et al. (2019) use cosine distance (equivalent to $d_{\mathbb{R}}^2$), while Wang et al. (MMA, 2021) use $d_{\mathbb{S}}$. Plugging in a log-distance instead of a distance gives the dispersion method proposed by Sablayrolles et al. (2019):

$$\mathcal{L}_{\text{KoLeo}}(X) = -\frac{1}{n} \sum_{i=1}^{n} \log \min_{j \neq i} d(x_i, x_j), \tag{4}$$

which is—up to a constant—the Kozachenko-Leonenko estimator of differential entropy (Leonenko, 1987). Assuming the distance is the same, the Kozachenko-Leonenko estimator and the max-min distance are related by the following bound which follows from the Jensen inequality and the fact that $-e^{-t} \leq t - 1$:

$$-d_{\min}(X) \leq \mathcal{L}_{\text{MM}}(X) \leq -\exp\left(-\mathcal{L}_{\text{KoLeo}}(X)\right) \leq \mathcal{L}_{\text{KoLeo}}(X) - 1. \tag{5}$$

We also note that technically the Kozachenko-Leonenko estimator is not applicable when the data is supported on a lower-dimensional submanifold such as $\mathbb{S}_m$: the differential entropy *w.r.t.* the Lebesgue measure on $\mathbb{R}^m$ is infinite. This issue is addressed in the estimator of Nilsson & Kleijn (2007). Nevertheless, as Eq. (5) suggests, seen simply as an optimization objective, $\mathcal{L}_{\text{KoLeo}}$ has the intended dispersion effect.

**Energy and kernel methods.** While the Tammes problem is defined in terms of the minimum distance, a related problem is to minimize an energy of the form:

$$\mathcal{L}_{\text{MHE},k}(X) = \frac{1}{n(n-1)} \sum_{1 \leq i \neq j < n} k(x_i, x_j). \tag{6}$$

MHE stands for *minimum hyperspherical energy* (Liu et al., 2018).[2] When $k$ is the Riesz 1-kernel with Euclidean distance, *i.e.*, $k(x, x') = d_{\mathbb{R}}(x, x')^{-1}$, $\mathcal{L}_{\text{MHE},k}(X)$ corresponds to minimizing the electrostatic energy between $n$ electrons, and its minimization is known as the Thomson problem (Thomson, 1904). This energy objective is minimized in a number of works on dispersion (Gautam & Vaintrob, 2013; Liu et al., 2018; 2021). Similarly, Wang & Isola (2020) propose a related objective using the RBF (also known as Gaussian) kernel $k(x_i, x_j) = \exp\left(\gamma \langle x_i, x_j \rangle\right)$ in the overall expression

$$\mathcal{L}_{\text{WI}}(X) = \log \frac{1}{n^2} \sum_{i=1}^{n} \sum_{j=1}^{n} k(x_i, x_j). \tag{7}$$

---

[1]In the case of ties, the function is not differentiable. Picking any closest pair ignoring ties gives a subgradient.
[2]Some definitions omit the scaling term $n(n-1)$.

Denoting $t(X) = \sum_i k(x_i, x_i)$, we thus have $\exp(\mathcal{L}_{\mathrm{WI}}(X)) = (1 + 1/n^2)(L_{\mathrm{MHE},k}(X) + t(X))$. For many kernels (including RBF, but not Riesz with $s \geq 0$), $t(X)$ is a finite constant independent on $X$ and thus, as exp is monotonic, the two objectives are optimization-equivalent. We shall revisit kernel properties in §3.1. While in general such energy minimization objectives lead to different optimal solutions, some relationships to the Tammes problem exist. For example, using the Riesz kernel with $s \to \infty$, or the RBF kernel with $\gamma \to \infty$, are equivalent to the Tammes problem in the limit (Appendix A.1).

Evaluating a kernel energy requires the entire (or at least the upper triangle of the) sample Gram matrix $K$ with $(K)_{ij} = k(x_i, x_j)$, so these methods have at least quadratic complexity, just like the closest-point methods above. Liu et al. (2021) also show that the determinant of $K$ is a dispersion measure; nevertheless calculating determinants require diagonalization, which has cubic complexity in $n$.

**Non-pairwise objectives.** Spherical variance (Eq. (2)) requires only linear time to evaluate. However, it is unfortunately not suitable for gradient-based optimization, as $\nabla_x \mathrm{svar}(X) = -x/2n\|\mu\|$ (it is in the direction of $x$), and thus $\mathrm{grad}_x \mathrm{svar}(X) = 0$ for any point $x$ part of any configuration $X$. Intuitively, svar wants to shrink all points toward the origin, which can result in no optimization progress on the sphere. As a generalization of spherical variance, it is possible to define more generic manifold notions of variance using Fréchet expectations, i.e. $\min_x \sum_i d(x, x_i)^2$ (Pennec, 2006), or variants using kernels instead of distances (maximum polarization, Liu et al., 2021). While avoiding explicit pairwise comparisons, such measures require iterative bilevel optimization, which poses additional numerical and computational challenges; moreover, for certain symmetrical optimally-dispersed situations any point on $\mathbb{S}_m$ solves the inner optimization and thus higher-order gradients can be ill-behaved.

**Optimal transport and spherical sliced Wasserstein.** Seeing dispersion as a semi-discrete distance between an empirical distribution $p$ (supported on a set of points $X$) and a uniform distribution $u$, it is tempting to turn to optimal transport (Villani et al., 2008; Peyré et al., 2019) and optimize $\mathcal{W}_2^2(p, u) :=$ $\min_{\pi \in \mathcal{U}(p,u)} \mathbb{E}_{(\mathsf{X},\mathsf{Y}) \sim \pi}\left[d(\mathsf{X}, \mathsf{Y})^2\right]$, where $\mathcal{U}(p, u)$ is the set of all joint measures with marginals matching $p$ and $u$. In most cases, including the case of the sphere, this Wasserstein distance is intractable, but (Bonet et al., 2023) show a closed-form special case on $\mathbb{S}_1$ with the geodesic distance. On the circle, a collection of points can be identified by their angular coordinates $\theta_1 \leq \theta_2 \leq \ldots \leq \theta_n$ with $-\pi \leq \theta_i \leq \pi$, and the Wasserstein distance against the uniform measure has the closed-form expression

$$W_2^2(p, u) = \frac{1}{n}\sum_{i=1}^{n}\theta_i^2 - \left(\frac{1}{n}\sum_{i=1}^{n}\theta_i\right)^2 + \frac{1}{n^2}\sum_{i=1}^{n}(n - 1 + 2i)\theta_i + \frac{1}{12} \tag{8}$$

Bonet et al. (2023) suggest the *spherical sliced Wasserstein* (SSW) distance, an efficient approximation over $\mathbb{S}_m$ by taking the expectation over the projections onto random great circles, similar to the more general idea of *sliced Wasserstein* in Euclidean spaces (Rabin et al., 2012), and they suggest that the distance to uniform may be used to optimize for dispersion. A great circle can be identified by a pair of orthogonal directions, and the projection is given by the following result.

> **Lemma 1 (Projection onto great circle.)** *Let $p, q \in \mathbb{S}_m$ with $\langle p, q \rangle = 0$. Two such vectors determine a unique great circle $\mathbb{S}_{pq} \subset \mathbb{S}_m$ defined by $\mathbb{S}_{pq} := \{\cos(\theta)p + \sin(\theta)q \mid -\pi \leq \theta < \pi\} \simeq \mathbb{S}_1$. The nearest point on $\mathbb{S}_{pq}$ to a given $x \in \mathbb{S}_m$ is given by $\mathrm{proj}_{\mathbb{S}_{pq}}(x) = \mathrm{arctan2}\left(\langle x, q \rangle, \langle x, p \rangle\right)$.*

As SSW was not experimentally compared with other dispersion strategies, we provide empirical results in our work. Moreover, in §3.1 we show that energy-based regularizers can be seen as optimizing a different distance between distributions: the MMD distance rather than the Wasserstein distance, and in §3.3 we use the slicing idea to derive from first principles a related but somewhat simpler regularizer with good performance.

## 3 New Insights Into Optimizing for Dispersion

In this section, we give a new interpretation of the uniformity regularizer discussed in §2, in terms of (squared) MMD, and then define two novel dispersion objectives with more appealing computational properties.

### 3.1 Energy-based dispersion and MMD

A promising way of thinking about dispersion comes from the statistical perspective of sample-based tests of uniformity. Indeed, spherical variance appears in the Rayleigh test for uniformity on the hypersphere $\mathbb{S}_m$ (Mardia & Jupp, 1999, 10.4.1), with test statistic $mn\|1 - \mathrm{svar}(X)\|^2$. Liu et al. (2018) show further connections between Ajne's test (Mardia & Jupp, 1999, 10.4.1) and MHE with a specific choice of kernel.

An alternative statistical test for uniformity can be derived from the *maximum mean discrepancy* (MMD), which is a nonparametric test measuring the distance between two probability distributions (Gretton et al., 2012). The next result shows that on $\mathbb{S}_m$, for kernels defined in terms of angles, the squared MMD from the uniform distribution simplifies to a familiar form.

> **Lemma 2** (MMD$^2$ **with the uniform distribution on** $\mathbb{S}_m$**.**) *Let $p$ be a distribution on $\mathbb{S}_m$, and denote by $u = \mathrm{Uniform}(\mathbb{S}_m)$ the uniform distribution on $\mathbb{S}_m$. Let $k$ be a positive definite kernel on $\mathbb{S}_m$ such that $k(x, y) = f(\langle x, y \rangle)$ for some function $f \colon [-1, 1] \to \mathbb{R}$, and $\mathcal{F}$ be the unit ball in the reproducing kernel Hilbert space (RKHS) corresponding to $k$.*
>
> *Then, for i.i.d. random variables $X, X'$ with distribution $p$, we have*
>
> $$\mathrm{MMD}^2[\mathcal{F}, p, u] = \mathbb{E}_{X, X' \sim p}\left[k(X, X')\right] - c,$$
>
> *where $c = B\left(\dfrac{1}{2}, \dfrac{m-1}{2}\right)^{-1} \displaystyle\int_{-1}^{1} dt\, f(t)(1 - t^2)^{(m-3)/2}$ is a constant w.r.t. $p$.*

While the constant is irrelevant for learning, we note that for the RBF $d_{\mathbb{R}}$ kernel, it takes the value of the Langevin distribution normalizing constant relative to the uniform measure, *i.e.*, $c = \Gamma(m/2)(\gamma/2)^{1-m/2}I_{m/2-1}(\gamma)$ (Mardia & Jupp, 1999, 9.3.6). The MMD is generally faster to approximate than the Wasserstein distance as there is no minimization over couplings. The assumption of expressing kernels in terms of the cosine $\langle x, y \rangle$ applies broadly and is required for rotational symmetry, and is analogous to expressing kernels in terms of distances, as the geodesic and Euclidean distances can both be given in terms of the cosine. The proof of Lemma 2 is provided in Appendix A. As a direct consequence, we may interpret $\mathcal{L}_{\mathrm{MHE},k} - c$ as an unbiased estimator of squared MMD against the uniform distribution on $\mathbb{S}_m$:

> **Proposition 1** *Let $p$ be any distribution on $\mathbb{S}_m$ and $k$ a positive definite kernel with the property from Lemma 2. Then, $\mathcal{L}_{\mathrm{MHE},k} - c$ as in Eq. (6) is an unbiased estimator of MMD$^2$ between $p$ and the uniform distribution on the sphere, with respect to the unit ball in the corresponding reproducible kernel Hilbert space.*

*Proof.* Combine Lemma 2 with Lemma 6 of Gretton et al. (2012) and identify with Eq. (6). ∎

Since the terms involving $u$ are handled in closed-form (or constant), this estimator also has more favorable variance properties compared to the standard MMD, which would be estimated by drawing samples from $u$. We provide a central limit theorem and expressions for the variance in Appendix A.3.

For our application, we take $p$ to be an empirical distribution supported at a finite number of points $X$. Since the uniform distribution is not in this family, MMD cannot reach zero, but we still have a meaningful notion of distance between $X$ and the continuous uniform distribution. This result provides a valid interpretation of kernel-based dispersion objectives like MHE and WI without requiring taking a limit toward infinitely many points, as needed in the motivation of Wang & Isola (2020).

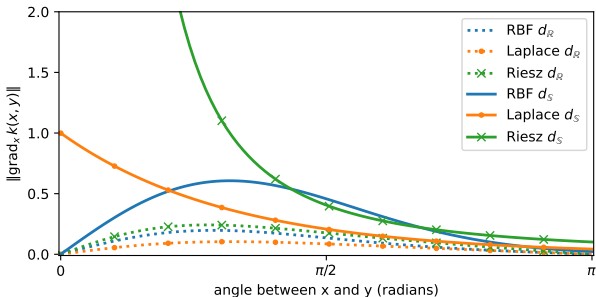

**Figure 2:** Norm of the gradient of the kernel between points, as the angle between points varies.

**Choice of kernel.** The most commonly-used kernels for dispersion are 1-Riesz $k_{\mathrm{Riesz},s=1}(x, y) = d(x, y)^{-1}$ (due to its connection to the Thomson problem), and

RBF $k_{\text{RBF},\gamma} = \exp(-\gamma d^2(x,x'))$. On the sphere (with $d_{\mathbb{S}}$), the RBF kernel is unfortunately not positive definite, while the Laplace kernel $k_{\text{Laplace},\gamma}(x,x') = \exp(-\gamma d(x,x'))$ is (Feragen et al., 2015). Inspired by Wang et al. (2021, Fig. 2), which shows the gradient norm of MMA is constantly 1 regardless of the angle to the nearest neighbor, we compare in Figure 2 the gradient norms for various kernels as a function of the angle to the other point. We find the geodesic Laplace kernel to be best-behaved from this perspective, as it is bounded and monotonically decreasing. All kernels that use $d_{\mathbb{R}}$, and also the RBF kernel with $d_{\mathbb{S}}$, first increase for small angles before decreasing. We give a detailed table of these kernels in Appendix A.1, alongside known results for positive definiteness. While this figure only depends on the angle and thus is the same in any dimension, the curse of dimensionality says that as dimension increases, angles converge to $\pi/2$. We explore the sensitivity (variance) of kernels in high dimensions in Appendix D.1.

### 3.2 Lloyd's Algorithm

An alternative formulation of dispersion can be reached by considering dispersion as *quantization* of a uniform measure. Quantization is the problem of approximating a given measure by an empirical measure supported at a few centers. Lloyd's algorithm (Lloyd, 1982), henceforth *Lloyd*, is a celebrated algorithm, introduced initially for quantization of uniform intervals in $\mathbb{R}$ for signal processing, but subsequently broadly applied, *e.g.*, in machine learning and graphics. It is an iterative algorithm that alternates between two stages. First, for each of the current centers, we find its *Voronoi cell*: the set of points in the domain closer to it than to the other centers. Afterward, all centers are replaced with the center of their corresponding Voronoi cell.

When the target measure is another empirical measure, quantization recovers *k-means clustering*; for continuous measures, it can be seen as an infinite generalization thereof. Both the discrete and continuous cases generalize readily to Riemannian manifolds with an adequate choice of distance (Le Brigant & Puechmorel, 2019). While Lloyd's algorithm and *k*-means are originally batch algorithms, stochastic gradient versions have been developed (Bottou & Bengio, 1995; Sculley, 2010), including, independently, in the Riemannian case (Le Brigant & Puechmorel, 2019). In general, given a domain $\mathbb{D}$, which could be a manifold or a compact subset of one (for quantization), or a discrete dataset (for clustering), the $n$ optimal centroids are a minimizer of[3]

$$\mathcal{L}_{\text{Lloyd}}(X) = \mathbb{E}_{\mathsf{Y} \sim \text{Unif}(\mathbb{D})} \left[ \min_{j \in [n]} \frac{1}{2} d^2(\mathsf{Y}, x_j) \right]. \tag{9}$$

A stochastic gradient of the Lloyd regularizer can be obtained by drawing $m$ uniform samples on $\mathbb{D}$. Intuitively, each cluster center is pulled toward the barycenter of the uniform samples assigned to it; an approximation to the true Voronoi barycenter.

For dispersion on the sphere, we take $\mathbb{D} = \mathbb{S}_m$ and $d = d_{\mathbb{S}}$. While traditionally Lloyd's algorithm corresponds to minimizing $\mathcal{L}_{\text{Lloyd}}$ alone, we propose using $\mathcal{L}_{\text{Lloyd}}$ as a regularizer to move $X$ closer to optimal Voronoi centers of the sphere, while also minimizing some main task-specific objective. The complexity of this regularizer is controlled by the number of samples: For efficiency, $m$ should be much less than $n$. Notice that unassigned cluster centers are not updated in an iteration, but the stochastic gradient of assigned centers implicitly depends on all other centers competitively through the cluster assignment. We additionally discuss convergence guarantees for Lloyd optimizer in Appendix C.

### 3.3 Sliced Dispersion

The previously discussed algorithms are generally applicable to other manifolds. We now show how using properties of the sphere we may obtain an alternative algorithm. We propose a construction similar to the idea used in spherical sliced Wasserstein, discussed in §2.4. While in three or more dimensions it is hard to find the location of $n$ evenly distributed points, on the $\mathbb{S}_2$ circle this can be done efficiently. The following set of angles is one optimal configuration:

$$\Phi = (\phi_1, \ldots, \phi_n) \quad \text{where} \quad \phi_k = -\pi \frac{n+1}{n} + \frac{2\pi k}{n}.$$

---

[3]The target measure need not be uniform. Le Brigant & Puechmorel (2019) give general conditions for existence.

Any other optimal configuration must be a rotation of this one, *i.e.*, $\tau + \Phi$ for $\tau \in (-\pi, \pi)$, followed by a permutation of these angles. Given a permutation $\sigma \in \Pi_n$ denote $\Phi_\sigma = (\phi_{\sigma(1)}, \ldots, \phi_{\sigma(n)})$. We can then write the set of all possible ordered optimally-dispersed configurations as

$$D_n\mathbb{S}_2 \coloneqq \{\tau + \Phi_\sigma \mid \tau \in (-\pi, \pi), \sigma \in \Pi_n\}. \qquad (10)$$

Given an ordered configuration of angles $\Theta = (\theta_1, \ldots, \theta_n) \subset \mathbb{S}_2$, we define its (angular) distance to the maximally-dispersed set as:

$$d^2(\Theta, D_n\mathbb{S}_2) = \min_{\hat{\Theta} \in D_n\mathbb{S}_2} \sum_{i=1}^{n} \frac{1}{2}(\theta_i - \hat{\theta}_i)^2. \qquad (11)$$

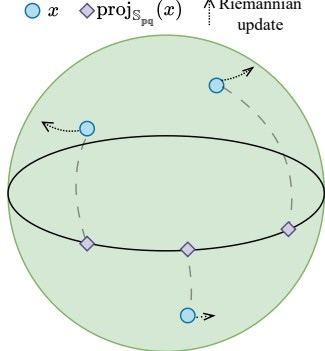

**Figure 3:** Sliced dispersion update along a great circle $\mathbb{S}_{pq}$.

Projecting onto $D_n\mathbb{S}_2$ means finding the closest dispersed configuration to a given one. The following lemma shows how to efficiently compute this projection, in the time it takes to sort $n$ angles.

**Lemma 3** *Optimal circular dispersion.* The projection $\arg\min_{\hat{\Theta} \in D_n\mathbb{S}_1} \sum_{i=1}^{n} 1/2 \left(\theta_i - \hat{\theta}_i\right)^2$ is given by $\hat{\theta}_i^\star = \tau^\star + \phi_{\sigma^{-1}(i)}$, where $\sigma$ is the permutation such that $\theta_{\sigma(1)} \leq \theta_{\sigma(2)} \leq \cdots \leq \theta_{\sigma(n)}$, and $\tau^\star = \sum_i \theta_i/n$.

In arbitrary dimensions, a similar construction is not possible, since the optimal configurations do not have tractable characterizations.

A well-dispersed configuration over $\mathbb{S}_m$ should remain fairly well-dispersed along any slice on average. If we denote $\text{proj}_{\mathbb{S}_{pq}}(X) \coloneqq (\text{proj}_{\mathbb{S}_{pq}}(x_1), \ldots \text{proj}_{\mathbb{S}_{pq}}(x_n))$, we may capture this intention by the following **sliced dispersion regularizer**:

$$\mathcal{L}_{\text{Sliced}}(X) = \mathbb{E}_{\mathsf{P},\mathsf{Q}}\left[d^2(\text{proj}_{\mathbb{S}_{\mathsf{PQ}}}(X), D_n\mathbb{S}_{\mathsf{PQ}})\right], \qquad (12)$$

where $d^2$ is defined in Eq. (11). The joint distribution of $\mathsf{P}$ and $\mathsf{Q}$ is a distribution over great circles. A sensible choice is the uniform distribution over great circles, which can be sampled by orthogonalizing a $2 \times m$ matrix with standard normal entries. An alternative is to sample from the discrete uniform distribution over axis-aligned great circles, *i.e.*, $p = e_i, q = e_j$ with $i, j$ drawn without replacement from $\{1, \ldots, m\}$; the latter choice allows slightly faster sampling and projection.

Note that unlike algorithms such as principal geodesic analysis (Fletcher et al., 2004), which keep $X$ fixed and optimize for $p, q$ to maximize variance, our intuition is the opposite: we update $X$ to increase dispersion along *all* great circles. The following proposition efficiently computes stochastic gradients of $\mathcal{L}_{\text{Sliced}}$.

**Proposition 2** *Denote $\theta_i|_{pq} = \text{proj}_{\mathbb{S}_{pq}}(x_i)$, and $\hat{\theta}_i^\star|_{pq}$ the corresponding dispersion maximizer computed using Lemma 3. The Euclidean and Riemannian gradients of $\mathcal{L}_{\text{Sliced}}$ are equal and given by:*

$$\text{grad}_{x_i} \mathcal{L}_{\text{Sliced}}(X) = \mathbb{E}_{P,Q}\left[\left(\theta_i|_{PQ} - \hat{\theta}_i^\star|_{PQ}\right) \frac{\langle x_i, P \rangle Q - \langle x_i, Q \rangle P}{\langle x_i, Q \rangle^2 + \langle x_i, P \rangle^2}\right]. \qquad (13)$$

All proofs in this section are provided in Appendix B.

Since the objective is defined as an expectation, applying stochastic Riemannian gradient is straightforward and we have an unbiased estimator. However, we remark that the standard convergence argument for stochastic Riemannian gradient does not hold for Sliced or for SSW, as the gradient not everywhere bounded. This shared issue is due to the great circle projection (Lemma 1), whose gradient (the fraction in Eq. (13)) tends to infinity whenever some $x_i$ tends toward the space orthogonal to the great circle. (The projection of the north pole to the equator is not uniquely defined). In high dimensions and/or low numerical precision this may trigger numerical issues, but we find that handling them heuristically (*e.g.*, via clipping) works well.

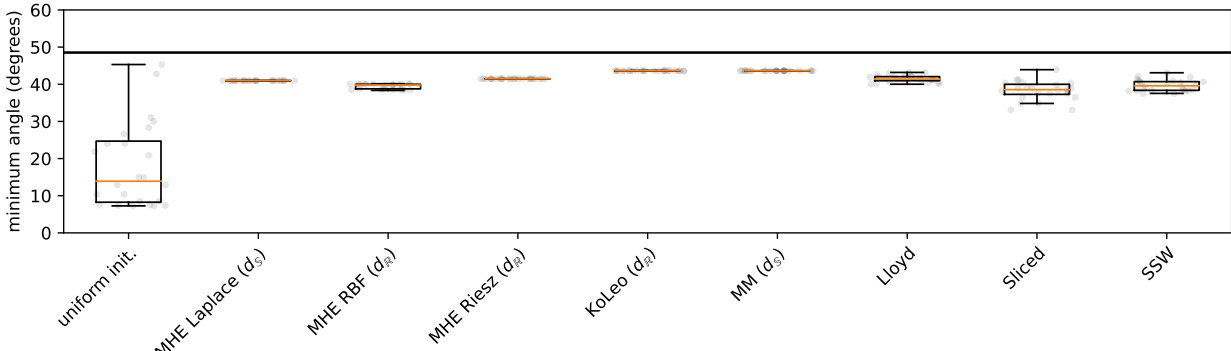

**Figure 4:** Minimum angles (degrees) for each of the $n = 24$ points. The horizontal line indicates the angle for known optimal solution; in an ideal solution, all angles would be on the horizontal line.

## 4 Applications

We demonstrate the application of dispersion objectives and provide a comparative analysis on both synthetic and real-world tasks.

### 4.1 Tammes problem

We evaluate the dispersion methods introduced in §2 and §3 by verifying that they can approximate the known solution to the Tammes problem for $n = 24$ in three dimensions (Robinson, 1961), by considering the minimum angle between points of the optimal configuration. Uniformly sampled points are dispersed using the regularizers described in §2 and §3. Starting from a uniform initialization, we apply 10k iterations of Riemannian Adam (Becigneul & Ganea, 2019) with learning rate 0.005. Laplace and RBF kernels use $\gamma = 1$, Riesz kernels use $s = 1$. Lloyd uses 300 samples from the uniform measure, number of projections for SSW is set to 50, while the sliced method uses a single random great circle per iteration.

The pairwise methods with min-max objective (MM and KoLeo) perform best as shown in Figure 4, unsurprisingly, as they are closest-related to the Tammes objective; they are followed by Lloyd and SSW/Sliced. Among the MHE methods, the Laplace kernel with geodesic distance works best here. For comparison, in Appendix E.3 we perform this experiment with Euclidean projected Adam instead of Riemannian Adam, finding a decrease in performance.

### 4.2 Synthetic Embeddings

In practice, we are mostly interested in dispersion of large amount of points in dimension $d \gg 3$. We evaluate the behavior of the regularizers discussed in §2 on $n = 20$k points in dimension $m = 64$, under two initial conditions: uniform samples (thus already somewhat spread out), and clumped points (sampled from a power spherical distribution with scale $\kappa = 100$) (De Cao & Aziz, 2020).

We optimize using Riemannian Adam for 5k iterations with learning rate 0.001. Laplace and RBF kernels use $\gamma = 1$, Riesz kernels use $s = 1$, and the Sliced regularizer uses axis-aligned great circles for efficiency. We set the batch size and number of samples to result in the same order of magnitude of computations: for all pairwise objectives (MHE, MM, KoLeo) we use minibatches of 512 points. For Lloyd, we also take minibatches of size 512, and draw 512 random uniform points to estimate the expectation. For Sliced we draw 13 random

| method | steps/sec ($\pm$ s.e.) |
|---|---|
| MHE (RBF, $d_{\mathbb{R}}$) | $79.40 \pm 0.06$ |
| KoLeo ($d_{\mathbb{R}}$) | $90.52 \pm 0.07$ |
| MM ($d_{\mathbb{S}}$) | $91.29 \pm 0.06$ |
| Lloyd | $90.98 \pm 0.07$ |
| Sliced | $87.72 \pm 0.14$ |
| SSW (nproj=13) | $53.78 \pm 0.25$ |
| SSW (nproj=1) | $81.16 \pm 0.12$ |

**Table 1:** Timing (mean gradient steps per second) and standard error of the mean for the synthetic experiments.

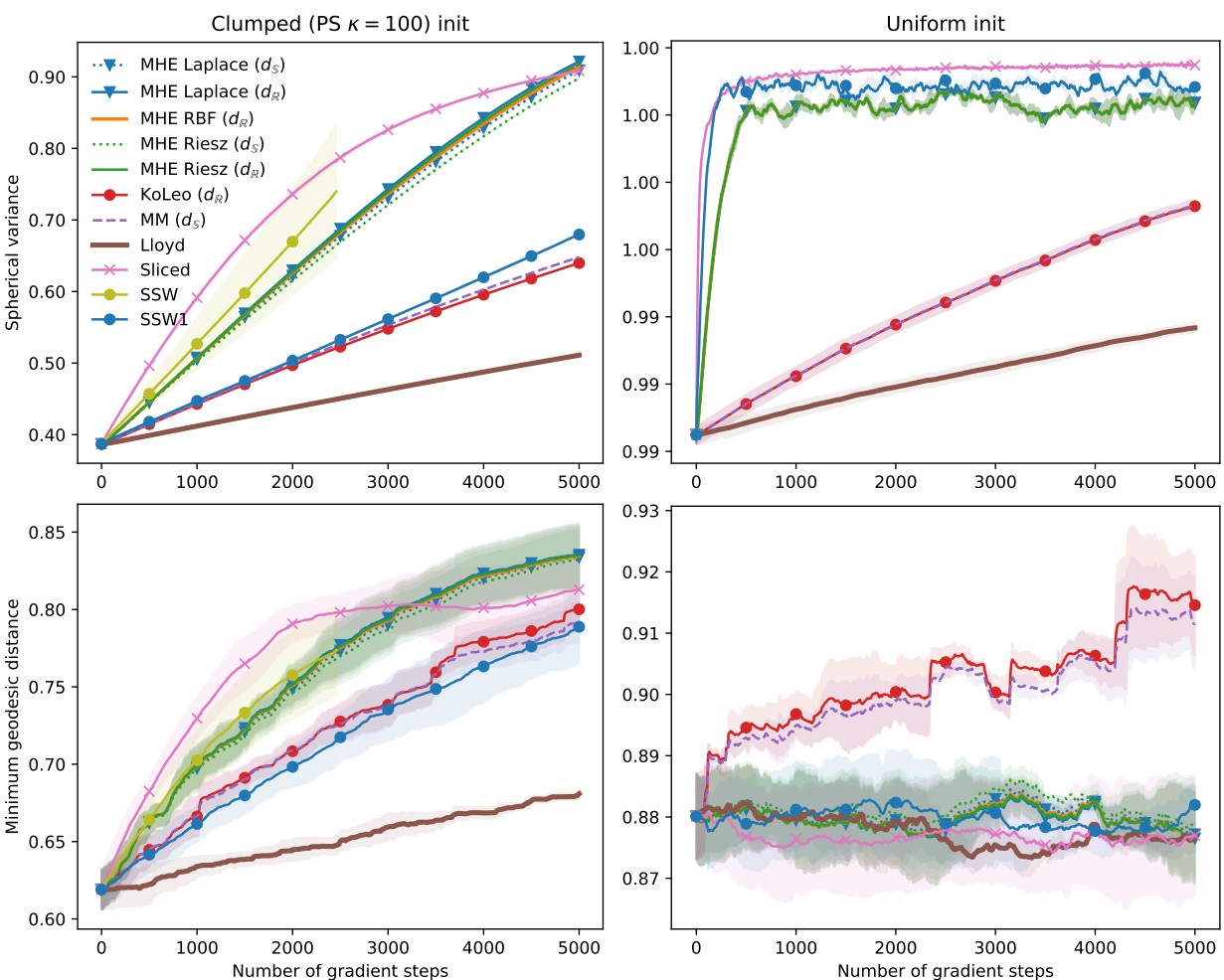

**Figure 5:** Convergence and performance of different dispersion objectives on synthetic data with $n = 20\,000, m = 64$. Mean and its standard error over three random initializations.

axis-aligned circles (as $13n \approx 512^2$). For SSW we provide results with 1 and 13 random circles, since the model with 13 random circles shows instability in the clumped case and diverges. We report spherical variance and minimum distances in Figure 5, and additionally measure timings in Table 1.

In contrast to the small-scale case of the Tammes problem, we find the choice of kernel and distance in MHE to not be empirically important in this large-scale regime. We find somewhat different trends in the clumped versus the uniform initialization scenarios. The clumped case roughly corresponds to a optimization alongside a strong downstream objective that pushes against dispersion. In this case, we find that MHE and Sliced perform best, with the latter making faster progress initially. In the uniform scenario, when the points are already well-separated, MHE and Sliced remain strong in terms of spherical variance, but the MM-style regularizers are the only ones that manage to make progress in terms of $d_{\min}$. In terms of speed (Table 1), even after balancing for amount of computation, we find MHE to be slower than the others.

Overall, these results suggest that Sliced is a contender for scenarios where initial dispersion is poor and fast progress is wanted, while MaxMin objectives seem preferable after some dispersion has already been achieved. We include additional results for varying dimensionality in D.2. Of course, $d_{\min}$ and svar do not necessarily correlate with improvement on downstream task metrics; the next sections are dedicated to such applications.

### 4.3 Image Classification with Prototypes

Mettes et al. (2019) showed that learning prototypes with dispersion encouraged by minimizing the maximum cosine similarity on a hypersphere improves classification results on ImageNet-200 (Le & Yang, 2015). In our naming system, this objective is $\mathcal{L}_{\mathrm{MM}}$ with $d_{\mathbb{R}}^2$. We first show in Table 2 that applying Riemannian

| prototypes | 50 | | 100 | | 200 | |
|---|---|---|---|---|---|---|
| | Acc. | $d_{\min}$ | Acc. | $d_{\min}$ | Acc. | $d_{\min}$ |
| MM ($d_{\mathbb{R}}^2$, projected) | $41.13_{\pm 0.34}$ | 1.22 | $43.39_{\pm 0.46}$ | 1.36 | $43.43_{\pm 0.28}$ | 1.44 |
| MM ($d_{\mathbb{R}}^2$) | $42.38_{\pm 0.16}$ | 1.46 | $43.27_{\pm 0.31}$ | 1.52 | $42.88_{\pm 0.29}$ | 1.56 |
| KoLeo ($d_{\mathbb{R}}$) | $41.72_{\pm 0.09}$ | 1.37 | $42.93_{\pm 0.23}$ | 1.44 | $42.55_{\pm 0.13}$ | 1.49 |
| MM ($d_{\mathbb{S}}$) | $42.11_{\pm 0.28}$ | 1.39 | $43.23_{\pm 0.17}$ | 1.46 | $42.73_{\pm 0.28}$ | 1.51 |
| MHE (Riesz, $d_{\mathbb{S}}$) | $42.98_{\pm 0.29}$ | 1.41 | $42.53_{\pm 0.22}$ | 1.56 | $33.70_{\pm 0.54}$ | 1.58 |
| MHE (RBF, $d_{\mathbb{R}}$) | $\mathbf{43.37}_{\pm 0.36}$ | 1.22 | $42.64_{\pm 0.09}$ | 1.57 | $33.36_{\pm 0.83}$ | 1.58 |
| Lloyd | $41.58_{\pm 0.08}$ | 1.20 | $42.43_{\pm 0.03}$ | 1.30 | $43.07_{\pm 0.40}$ | 1.35 |
| Sliced | $40.91_{\pm 0.11}$ | 1.10 | $42.68_{\pm 0.45}$ | 1.20 | $42.62_{\pm 0.30}$ | 1.33 |
| SSW | $40.44_{\pm 0.31}$ | 1.08 | $42.34_{\pm 0.30}$ | 1.18 | $43.01_{\pm 0.22}$ | 1.29 |

**Table 2:** ImageNet-200 classification accuracy. Prototypes are trained with different separation conditions. In bold we emphasise the best accuracy in a column. MM ($d_{\mathbb{R}}^2$, projected) is the objective of Mettes et al. (2019) optimized with Euclidean projected gradient; all other rows use Riemannian optimization.

gradient descent on the sphere, rather than projected gradient as done by Mettes et al. (2019), leads to the better class separation, and as a result better classification accuracy. Second, we compare the classification accuracy given the prototypes trained with different dispersion objectives discussed in §2 and §3. We use Riemannian gradient descent for all except the projection baseline; we use $\gamma = 1$ for RBF and $s = 1$ for Riesz kernels, 200 uniform samples in Lloyd, and a single sampled great circle for Sliced. Also, Table 2 shows that, for $m = 50$, MHE with the RBF kernel performs the best among all dispersion objectives considered, even though the minimum distance is smaller compared to other pairwise-distance based objectives. It proves that even though we can measure the dispersion using minimum distance, we cannot rely on this metric alone as a predictive factor of the downstream task accuracy.

Interestingly, when dimensionality is equal to the number of points, MHE results degrade noticeably, as shown in Table 2. For both kernels considered, the minimum distance and median distance is approximately 1.57 radian or 90.3°, which is close to an orthogonal solution. Orthogonality is suggested by Mettes et al. (2019) to be insufficient despite being optimally dispersed.

### 4.4 Neural Machine Translation

Embeddings learned with the vanilla transformer model (Vaswani et al., 2017) are known for their inefficiency in utilizing space effectively, leading to the collapse of token representations (Gao et al., 2019; Wang et al., 2020). This issue is particularly pronounced for rare tokens (Gong et al., 2018; Tokarchuk & Niculae, 2024; Zhang et al., 2022). Gong et al. (2018) proposed to alleviate the problem of rare tokens by learning frequency-agnostic embeddings, while Zhang et al. (2022) proposed to use contrastive learning. In our approach, we tackle this challenge by focusing on the concept of dispersion. Specifically, we train a neural machine translation (NMT) system and optimize the transformer weights $W$ along with a dispersion regularizer on the decoder embeddings $X$ (tied between the input and the output decoder layers):

$$\mathcal{L}(W, X) = \mathcal{L}_{\mathrm{MT}}(W, X) + \lambda \mathcal{L}_{\mathrm{disp}}(X), \tag{14}$$

where $\mathcal{L}_{\mathrm{MT}}$ is a standard conditional log-likelihood loss of the target sentence given the source sentence accumulated over a dataset (*e.g.*, Bahdanau et al., 2015). We perform stochastic training: for every $\mathcal{L}_{\mathrm{MT}}$ minibatch gradient calculation, we calculate an (independent) stochastic gradient of the regularizer. In particular, for MHE, we estimate the dispersion gradient over a minibatch of 1000 tokens drawn from the whole vocabulary. For Lloyd we use 100 uniform samples from the sphere, while for Sliced we sample one

| model | ro-en | | | | en-de | | | |
|---|---|---|---|---|---|---|---|---|
| | BLEU | COMET | $d_{\min}$ | svar | BLEU | COMET | $d_{\min}$ | svar |
| Euclidean baseline | 31.5 | 0.790 | 0.003 | 0.19 | 33.1 | 0.819 | 0.000 | 0.000 |
| spherical baseline | 32.2 * | 0.793 * | 0.001 | 0.57 | 33.7 * | **0.826** * | 0.001 | 0.408 |
| +MHE (RBF, $d_{\mathbb{R}}$) | 32.3 * | **0.795** * | 0.001 | 0.56 | **33.9** * | 0.825 * | 0.001 | 0.410 |
| +Lloyd | **32.4** * | 0.791 | 0.001 | 0.60 | 33.4 * | 0.822 | 0.001 | 0.414 |
| +Sliced | 32.3 * | **0.795** * | 0.435 | 0.99 | 33.5 * | 0.821 * | 0.222 | 0.999 |

**Table 3:** newstest2016 `ro-en` and `en-de` results on discrete NMT. Embeddings are 128 dim. All results marked with * are statistically significant ($p < 0.05$) over baseline.

| embeddings | svar ↑ | $d_{\min}$ ↑ | BLEU↑ |
|---|---|---|---|
| pretrained $\mathbb{R}^m$ (normalized) | 0.191 | 0.014 | 28.3 |
| +offline MHE (RBF, $d_{\mathbb{R}}$) | 0.599 | 0.372 | 29.7 * |
| +offline Lloyd | 0.585 | 0.004 | 27.7 |
| +offline Sliced | 0.979 | 0.106 | 29.6 * |
| pretrained $\mathbb{S}_m$ | 0.573 | 0.001 | 29.9 * |
| +MHE (RBF, $d_{\mathbb{R}}$) | 0.561 | 0.001 | 30.0 * |
| +Lloyd | 0.596 | 0.001 | **30.1** * |
| +Sliced | **0.999** | **0.435** | 30.0 * |

**Table 4:** Dispersion impact on CoNMT results. We report BLEU scores on the `newstest2016` for `ro-en`. Beam size is equal to 5. All results marked with * are statistically significant ($p < 0.05$) over baseline.

great circle, with no additional subsampling. We use $\gamma = 1$ for RBF. The remaining NMT training settings are standard and provided in Appendix E.1. We report results on two WMT translation tasks:[4] WMT 2016 Romanian→English (`ro-en`) with 612K training samples and WMT 2019 English→German (`en-de`) with 9.1M training samples (including back-translated data). We measure translation accuracy on the best checkpoint according to validation BLEU score using SacreBLEU (Papineni et al., 2002; Post, 2018) and COMET (Rei et al., 2020).

Table 3 shows the BLEU and COMET results on newstest2016 for `ro-en` and newstest2016 `en-de` along with the dispersion metrics. Surprisingly, simply training the transformer with Riemannian gradient updates for $X$ (and standard Euclidean gradients for all other parameters) alone already improves dispersion and translation performance over the baseline. Additional dispersion regularization further improves dispersion and BLEU, but not always COMET.

To shed more light into the dispersion effect of Riemannian optimization, we plot the relationship between token frequency and gradient norm (Figure 6a) / minimum pairwise distance (Figure 6b) in the late stage of training. While in the Euclidean model the embeddings of rare words can collapse close to their neighbor, the Riemannian model prevents this. The gradient norm of rare words is an order of magnitude higher with Riemannian gradients than with Euclidean gradients. We hypothesize that this increased gradient norm contributes to better dispersion of rare tokens, thereby mitigating representation collapse. The training dynamics of gradient norms and minimum distances along the earlier training stages, shown in Appendix E.2, support this intuition.

## 4.5 Continuous-Output Neural Machine Translation

Continuous-output NMT (CoNMT, Kumar & Tsvetkov, 2019) reformulates machine translation as a sequential continuous regression problem of predicting the embedding of the next word, instead of the more usual discrete classification formulation. Tokarchuk & Niculae (2024) recently showed that dispersion plays an important role and greatly impacts performance. We follow closely their setup and apply the dispersion

---

[4]https://www2.statmt.org/

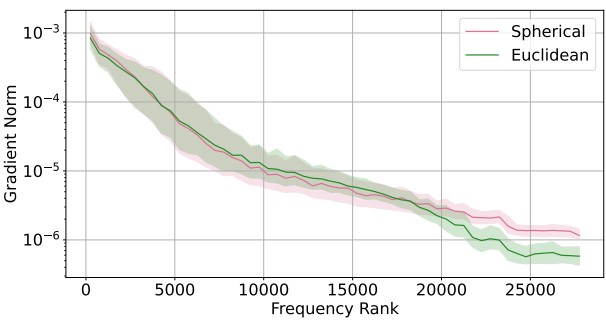 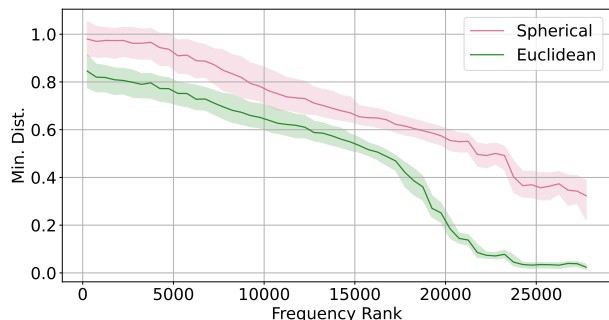

**(a)** Norm of the gradient of the embedding.

**(b)** Minimum $d_{\mathbb{S}}$ distance to the nearest embedding.

**Figure 6:** Gradient norms (a) and minimum distances (b) *w.r.t.* NMT embeddings, after 40k steps, for the standard (Euclidean) transformer and the spherical version where decoder embeddings are optimized on $\mathbb{S}_m$. Frequency rank is the index of the token in a sorted vocabulary: higher means less frequent.

regularizers in order to achieve dispersion. Pre-trained embeddings come from the well-trained discrete models from §4.4 and are frozen while the continuous model is trained from scratch. We present results for WMT 2016 `ro-en` with 612k training samples. Table 4 shows the BLEU score results on `newstest2016` for CoNMT models with different target embeddings $X$, alongside dispersion measures defined in §2.3

As baselines, we include (normalized) target embeddings from the best Euclidean discrete model (labeled *pretrained* $\mathbb{R}^m$). For *offline* dispersion we optimize starting from the pretrained embeddings for 500 epochs, using batch size 2000 in MHE, RBF $\gamma = 1$, 100 uniform samples in Lloyd, and a single sampled great circle for Sliced. Second, as we have already trained discrete NMT models with spherical dispersion in §4.4, we use the resulting embeddings for CoNMT (rows labeled *pretrained* $\mathbb{S}_m$). All CoNMT models were trained for 50k steps. Spreading out the projected embeddings generally results in BLEU score improvements. For all dispersion regularizers, we can see that svar is increasing. However, $d_{\min}$ decreases for Lloyd, which may explain the lower BLEU score. Consistent with the synthetic results (§4.2), the Sliced regularizer is best at quickly increasing svar.

### 4.6 Discussion

Pairwise objectives are a good choice when the full matrix of pairwise distances can be calculated efficiently, *i.e.*, when the number of datapoints and/or dimensionality are relatively small. In practical ML applications, dimensionality and number of points are typically large, as in §4.4 and §4.5, and given the limitations of computational resources we can either calculate pairwise distance for random batch of datapoints or use alternatives like Lloyd and Sliced, since they are more computationally efficient as shown in Table 5.

## 5 Additional Related Work

A prominent related but distinct construction is that of **determinantal point processes** (Kulesza & Taskar, 2012), often used to ensure diversity or dispersion in extractive models. Given a set of $n$ points, in the scope of our paper we study optimizing the locations of the points; in contrast, finite DPPs are intended for selecting subsets of $k \leq n$ such points in a way that encourages dispersion of the selected points. Technically, DPPs induce a probability distribution over all subsets. Like the pairwise regularizers we study in §2, DPPs form an $n \times n$ kernel matrix of pairwise similarities. In order to draw samples, or to evaluate the probability of a given subset, an eigendecomposition (or equivalent)

| method | complexity |
|---|---|
| pairwise | $\mathcal{O}(n^2 m)$ |
| Lloyd | $\mathcal{O}(nm)$ |
| Sliced / SSW | $\mathcal{O}(nm + \text{sort}(n))$ |
| (axis-aligned) | $\mathcal{O}(n + \text{sort}(n))$ |

**Table 5:** Computational complexity of the dispersion regularizers for $n$ samples in $\mathbb{S}_m$. For Lloyd, Sliced, and SSW the complexities are *w.r.t.* a single Monte Carlo sample and complexity scales linearly in case of multiple samples. $\text{sort}(n)$ is the complexity of sorting a list of length $n$, *i.e.*, $\mathcal{O}(n)$ with radix sort and $\mathcal{O}(n \log n)$ with comparison sorts.

of the kernel matrix is necessary, leading to $O(n^3)$ time complexity for such operations. Maximization, *i.e.*, finding an optimally-dispersed subset under a DPP, is NP-hard to solve exactly, but approximations may be worth exploring. Continuous DPPs can be defined on compact sets (Lavancier et al., 2014), including on $\mathbb{S}_m$ (Møller et al., 2018). Relatedly, in numerical integration it is essential to find good quadratures or discretizations of continuous measures. Methods include Sobol sequences (Sobol, 1967), defined on unit hybercubes, and kernel herding, for arbitrary measures (Bach et al., 2012). Particle systems such as Stein variational gradient descent Liu & Wang (2016); Liu & Zhu (2018) can be used to sample from uniform measures. Sisouk et al. (2025) discuss various strategies for sampling on $\mathbb{S}_m$ while ensuring good coverage. They show that orthonormal sampling (Rowland et al., 2019) specifically is the best in high-dimensional scenario. However, sampling is not directly applicable in a regularization application where a task-specific objective is also present. Nevertheless, further connections in this direction seem worth developing.

Dispersion is also closely connected to **contrastive learning** (Chen et al., 2020a; He et al., 2020; Hjelm et al., 2019; Chen et al., 2020b), where model outputs corresponding to different classes are pushed away from each other. Wang & Isola (2020) in particularly showed that widely used contrastive learning objective can be interpreted in terms of "alignment" (similar features for similar samples) and "uniformity" (feature distribution is close to uniform distribution). In our work we focus on parameter dispersion, which can more easily be quantified.

Marbut et al. (2023) explore other ways to quantify dispersion in Euclidean space, and studied their consequences for word embeddings. Exploring spherical extensions thereof is a direction for future work.

## 6 Conclusion

In this work, we evaluate several dispersion objectives on the hypersphere, including one that is equivalent to the widely used maximum mean discrepancy (MMD) method, as well as two novel approaches: Lloyd and Sliced. We compare these objectives against various pairwise distance-based methods previously explored in the literature. Our experimental results show that these methods can approximate the Tammes problem solution, and also allow improvement on few-shot image classification with prototypes, machine translation and the CoNMT tasks, which uses cosine distance both for training and decoding.

### Acknowledgments

This work is supported by the Dutch Research Council (NWO) via VI.Veni.212.228. The authors also thank SURF (www.surf.nl) for the support in using the National Supercomputer Snellius. Finally, we thank all the members of the UvA Language Technology Lab for their valuable feedback on our work, with special thanks to Sergey Troshin for his input on the manuscript.

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

# Appendix

## A    Kernels and MMD Dispersion

### A.1    Spherical Kernels and Relationships

In this section we give a convenient table of the expressions of the kernels used in this paper (Table 6), and additionally show the limit-case relationship to the Tammes problem.

| name | $d$ | hyperpar. | expression $k(x,y) =$ | (conditionally) positive definite |
|---|---|---|---|---|
| RBF | $\mathbb{R}$ | $\gamma \in (0,\infty)$ | $\exp\left(\gamma\langle x,y\rangle\right)$ | yes |
| RBF | $\mathbb{S}$ | $\gamma \in (0,\infty)$ | $\exp\left(-\gamma(\arccos\langle x,y\rangle)^2\right)$ | no (Feragen et al., 2015) |
| Laplace | $\mathbb{R}$ | $\gamma \in (0,\infty)$ | $\exp\left(-\gamma\sqrt{2-\langle x,y\rangle}\right)$ | yes |
| Laplace | $\mathbb{S}$ | $\gamma \in (0,\infty)$ | $\exp\left(-\gamma\arccos\langle x,y\rangle\right)$ | yes (Feragen et al., 2015) |
| Riesz | $\mathbb{S}$ | $s \in (0,\infty)$ | $(\arccos\langle x,y\rangle)^{-s}$ | cond., for $s < m-1$ (Bilyk & Grabner, 2024, Cor. 2) |
| | | $s = 0$ | $-\log\arccos\langle x,y\rangle$ | cond., (Bilyk & Grabner, 2024, Cor. 2) |
| Riesz | $\mathbb{R}$ | $s \in (0,\infty)$ | $(2-\langle x,y\rangle)^{-s/2}$ | cond., for $s < m-1$ (Bilyk & Grabner, 2024, Cor. 1) |
| | | $s = 0$ | $-1/2\log(2-\langle x,y\rangle)$ | cond., (Bilyk & Grabner, 2024, Cor. 1) |

**Table 6:** Expressions of the kernels considered in this work and in related work on hyperspherical dispersion. In all cases, the domain is a sphere $\mathbb{S}^m$. The Euclidean-distance RBF kernel is up to a constant equal to the one in ambient space. Riesz kernels for negative $s$ are sometimes defined as $k_{\mathrm{Riesz},s}(x,y) = -k_{\mathrm{Riesz},-s}(x,y)$.

**Limit cases.**    First, let's consider the RBF and Laplace kernels. For simplicity let $R$ denote the set of pairs of distinct indices from 1 to $n$, and $z_r$ be the cosine similarity between the $r$th pair of points in $X$. Then, the geodesic objective of the Tammes problem can be written as $d_{\min}(X) = \arccos\left(\max_{r\in R} z_r\right)$ and since arccos is nondecreasing, the optima of the Tammes problem thus also maximize $\max_{r\in R} z_r$. The "soft maximum" operator log-sum-exp satisfies a well-known limit property (Salmon & Collin, 2024; Asadi & Littman, 2017). Denoting $z_* = \max_{r\in R} z_r$ and assuming no ties,

$$
\lim_{\gamma\to\infty} \gamma^{-1}\log\sum_{r\in R}\exp(\gamma z_r) = \lim_{\gamma\to\infty}\gamma^{-1}\log\sum_{r\in R}\exp(\gamma(z_r - z_* + z_*))
$$

$$
= \lim_{\gamma\to\infty}\gamma^{-1}\log\left(\exp(\gamma z_*)\sum_{r\in R}\exp(\gamma\underbrace{(z_r - z_*)}_{<0})\right) \tag{15}
$$

$$
= z_*.
$$

Since log-sum-exp is continuous, in presence of ties, the same limit holds as the ties can be broken by some $\varepsilon$ going to zero. Introducing or not a normalization of $\frac{1}{|R|}$ before the sum does not change this result, as $\gamma^{-1}\log Z/|R| = \gamma^{-1}\log Z - \gamma^{-1}\log|R|$ and the latter term goes to zero in the limit. Moreover, any strictly monotonic transformation of the cosine similarities also preserves the optima, which means all versions of the RBF and Laplace kernels shown in Table 6 are equivalent to Tammes in the limit of $\gamma \to \infty$. The limit case of the Riesz kernel as $s \to \infty$ can be proven similarly and is related to the definition of the $\|\cdot\|_\infty$ norm. Let's redefine now $z_r = 1/d_r$ be the inverse of a distance (so still a similarity):

$$
\lim_{s\to\infty}\left(\sum_{r\in R}z_r^s\right)^{1/s} = z_* \lim_{s\to\infty}\left(1 + \sum_{r\in R}\underbrace{\left(\frac{z_r}{z_*}\right)^s}_{<1}\right)^{1/s} = z_*. \tag{16}
$$

The MHE regularizer with Riesz kernel as $s \to \infty$ is unbounded, but its $s$th root converges to an objective equivalent to the Tammes problem.

## A.2 $\mathrm{MMD}^2$ and spherical uniform densities: Proof of Lemma 2

The squared MMD of two probability distributions $p$ and $q$ is equal to (Gretton et al., 2012, Lemma 6)

$$\mathrm{MMD}^2[p, q] = \mathbb{E}_{\mathsf{X}, \mathsf{X}' \sim p}[k(\mathsf{X}, \mathsf{X}')] - 2\mathbb{E}_{\mathsf{X} \sim p, \mathsf{Y} \sim q}[k(\mathsf{X}, \mathsf{Y})] + \mathbb{E}_{\mathsf{Y}, \mathsf{Y}' \sim q}[k(\mathsf{Y}, \mathsf{Y}')].$$

We show that the last two expectations are constant, when $p$ is a distribution on the hypersphere $\mathbb{S}_m$ and $q$ is $\mathrm{Unif}(\mathbb{S}_m)$. Let $z, z' \in \mathbb{S}_m$ and let $Q$ be a rotation matrix such that $Qz = z'$. Note that $\mathsf{Y} \sim \mathrm{Unif}(\mathbb{S}_m)$ if and only if $Q^\top \mathsf{Y} \sim \mathrm{Unif}(\mathbb{S}_m)$, and $\langle Qz, z \rangle = \langle z, Q^\top z \rangle$. It then follows that

$$\mathbb{E}_{\mathsf{Y} \sim \mathrm{Unif}(\mathbb{S}_m)}[k(z, \mathsf{Y})] = \mathbb{E}_{\mathsf{Y} \sim \mathrm{Unif}(\mathbb{S}_m)}[k(z', \mathsf{Y})],$$

since $k(x, y) = f(\langle x, y \rangle)$. Hence, there exists a $c \in \mathbb{R}$ such that for all $z \in \mathbb{S}_m$ we have

$$\mathbb{E}_{\mathsf{Y} \sim \mathrm{Unif}(\mathbb{S}_m)}[k(z, \mathsf{Y})] = c.$$

Consequently, $\mathbb{E}_{\mathsf{X} \sim p, \mathsf{Y} \sim \mathrm{Unif}(\mathbb{S}_m)}[k(\mathsf{X}, \mathsf{Y})] = c$ and $\mathbb{E}_{\mathsf{Y}, \mathsf{Y}' \sim \mathrm{Unif}(\mathbb{S}_m)}[k(\mathsf{Y}, \mathsf{Y}')] = c$.

To work out an expression for $c$ we use a standard strategy used, *e.g.*, by Mardia & Jupp (1999, 9.3.1). We reparametrize the sphere as $(t, u) \mapsto tz + \sqrt{1 - t^2}u$ for $t \in [-1, 1]$ and $u$ on the unit sphere in the space orthogonal to $z$ (*i.e.*, $\|u\| = 1$ and $\langle u, z \rangle = 0$). This reparametrization has the essential property that $f(\langle z, y \rangle) = f(t)$, allowing us to eventually reduce to a 1d integral. The domain of $u$ is isomorphic to $\mathbb{S}_{m-1}$, which is easiest to see if we take $z = (0, \ldots, 0, 1)$ which gives $y = [\sqrt{1 - t^2}\bar{u}, t]$ for $\bar{u} \in \mathbb{S}_{m-1}$. Putting everything together gives

$$c = \frac{1}{\mathrm{Surf}(\mathbb{S}_m)} \int_{\mathbb{S}_{m-1}} \mathrm{d}u \int_{-1}^{1} \mathrm{d}t f(t)(\sqrt{1 - t^2})^{m-3} = B\left(\frac{1}{2}, \frac{m-1}{2}\right)^{-1} \int_{-1}^{1} \mathrm{d}t f(t)(\sqrt{1 - t^2})^{m-3}, \quad (17)$$

where $B$ is the Beta function and we used the fact that $\mathrm{Surf}(\mathbb{S}_m)/\mathrm{Surf}(\mathbb{S}_{m-1}) = B(1/2, (m-1)/2)$. In particular, for $f(t) = \exp(\gamma t)$ this recovers the Langevin distribution normalizing integral as derived by Mardia & Jupp (1999, 9.3.6).

## A.3 Variance for a generic positive-definite kernel

Since the MMD estimator in Lemma 2 treats the terms involving $u$ as a constant and thus does not need to estimation, limits and bounds for variance can be better derived. In particular, because U-statistic (Hoeffding, 1963) is in terms of $k$ itself, Theorem 10 from (Gretton et al., 2012) can be modified as follows:

$$\Pr[\mathrm{MMD}_u^2(\mathcal{F}, p, u) - \mathrm{MMD}^2(\mathcal{F}, p, u) > t] \leq \exp\left(\frac{-t^2 \lfloor n/2 \rfloor}{K^2}\right)$$

*i.e.*, the denominator inside the exp is smaller by a factor of 8 compared to the formulation of (Gretton et al., 2012), and thus bounded in $[0, K]$. In terms of variance, we can use the expressions of Serfling (2009) as in (Bounliphone et al., 2015) and (Sutherland, 2019) (as well as the CLT in (Gretton et al., 2012) Corollary 16) to obtain the asymptotic expression

$$\mathrm{Var}[\mathrm{MMD}_u^2(\mathcal{F}, p, u)] = \frac{4(n-2)}{n(n-1)}\left(E[\langle \varphi(x), \mu_X \rangle^2] - \|\mu_X\|^2\right) + O(n^{-2})$$

or the similar second-order expression. These expressions are much simpler than the full MMD case since only the $p$ terms are needed, and the terms involving $u$ are constants. Note that in case of dispersion, which is our primary study focus, $p$ is an empirical measure and thus (with full-batch optimization) our regularizer is exact, but these variance results could be used to analyze the variance of the minibatch estimator.

# B Sliced Dispersion: Proofs

## B.1 Optimal Circle Dispersion

We aim to prove the assertion that the projection

$$\arg\min_{\hat{\Theta} \in D_n \mathbb{S}_2} \sum_{i=1}^{n} \frac{1}{2}(\theta_i - \hat{\theta}_i)^2$$

is given by $\hat{\theta}_i^{\star} = \tau^{\star} + \phi_{\sigma^{-1}(i)}$, where $\sigma$ is the permutation such that $\theta_{\sigma(1)} \leq \theta_{\sigma(2)} \leq \cdots \leq \theta_{\sigma(n)}$, and $\tau^{\star} = \frac{\sum_i \theta_i}{n}$.

By definition, per Eq. (10), $\hat{\Theta} = \tau + \Phi_\sigma$ and thus we may write the problem equivalently as

$$\arg\min_{\tau \in [-\pi, \pi), \sigma \in \Pi_n} \sum_i \frac{1}{2}(\theta_i - \phi_{\sigma(i)} - \tau)^2.$$

**Finding the permutation.** In terms of $\sigma$ the objective takes the form $-\sum_i \theta_i \phi_{\sigma(i)} + \text{const}$, so we must find the permutation that maximizes $\sum_i \theta_i \phi_{\sigma(i)} = \sum_i \theta_{\sigma^{-1}(i)} \phi_i$. By the rearrangement inequality (Hardy et al., 1952, Thms. 368–369), since $\phi_i$ is in ascending order, this sum is maximized when $\theta_{\sigma^{-1}(i)}$ is in ascending order; so the optimal $\sigma$ must be the inverse of the permutation that sorts $\Theta$.

**Finding $\tau$.** Ignore the constraints momentarily, and set the gradient of the objective to zero:

$$\frac{\partial}{\partial \tau} \sum_i \frac{1}{2}(\theta_i - \phi_{\sigma(i)} - \tau)^2 = \sum_i (\tau + \phi_{\sigma(i)} - \theta_i) = 0, \quad \text{implying} \quad n\tau = \sum_i \theta_i - \sum_i \phi_i = \sum_i \theta_i,$$

the last equality by choice of the zero-centered reference configuration $\Phi$. Since all $\theta_i \in [-\pi, \pi)$, so is their average, and thus the constraints are satisfied, concluding the proof.

## B.2 Projection onto a great circle

The projection we seek to compute is

$$\text{proj}_{\mathbb{S}_{pq}}(x) := \arg\min_{-\pi \leq \theta < \pi} d_{\mathbb{S}}^2(\cos(\theta)p + \sin(\theta)q, x).$$

Since the geodesic distance satisfies $d_{\mathbb{S}}(\cdot, \cdot) = \arccos\langle \cdot, \cdot \rangle$ and arccos is strictly decreasing on $(-1, 1)$, we have

$$\text{proj}_{\mathbb{S}_{pq}}(x) := \arg\max_{-\pi \leq \theta < \pi} \langle \cos(\theta)p + \sin(\theta)q, x \rangle.$$

As a side note, this shows that it doesn't matter whether we use geodesic or Euclidean distance to define this projection. Setting the gradient to zero yields

$$\cos(\theta)\langle q, x \rangle = \sin(\theta)\langle p, x \rangle,$$

or equivalently $\tan(\theta) = \langle q, x \rangle / \langle p, x \rangle$. The unique solution on $[-\pi, \pi)$ is given by the two-argument arctangent function (arctan2), also known as the argument of complex number $\langle p, x \rangle + i\langle q, x \rangle$ (Wikipedia contributors, 2024).

## B.3 Gradient of sliced distance

We first compute the Euclidean gradient of the desired expression:

$$\nabla_{x_i} \mathcal{L}_{\text{Sliced}}(X) = \nabla_{x_i} \mathbb{E}_{p,q} \left[ d_{\mathbb{S}}^2(\text{proj}_{\mathbb{S}_{pq}}(X), D_n \mathbb{S}_{pq}) \right]. \tag{18}$$

First, by writing

$$d_{\mathbb{S}}^2(\Theta, D_n \mathbb{S}_{pq}) = \min_{\hat{\Theta}} \sum_i \frac{1}{2}(\theta_i - \hat{\theta}_i)^2$$

we see this may be interpreted as a Euclidean projection and

$$\frac{\partial}{\partial \theta_i} d_{\mathbb{S}}^2(\Theta, D_n \mathbb{S}_{pq}) = (\theta_i - \theta_i^\star).$$

But $\theta_i = \text{proj}_{\mathbb{S}_{pq}}(x_i)$ and we can write

$$\begin{aligned}
\frac{\partial \theta_i}{\partial x_i} &= \frac{\partial}{\partial x_i} \text{proj}_{\mathbb{S}_{p,q}}(x_i) \\
&= \frac{\partial \theta_i}{\partial x_i} \tan^{-1}\left(\frac{\langle q, x \rangle}{\langle p, x \rangle}\right) \\
&= \frac{\langle p, x \rangle q - \langle q, x \rangle p}{\langle q, x \rangle^2 + \langle p, x \rangle^2}.
\end{aligned}$$

Putting the two together via the chain rule yields

$$\nabla_{x_i} \mathcal{L}_{\text{Sliced}}(X) = \left(\theta_i\big|_{pq} - \hat{\theta}_i\big|_{pq}\right) \frac{\langle p, x_i \rangle q - \langle q, x_i \rangle p}{\langle q, x_i \rangle^2 + \langle p, x_i \rangle^2}. \tag{19}$$

Notice that the second term is a vector in $\mathbb{R}^m$ that is orthogonal to $x_i$ because:

$$\langle x_i, \langle p, x_i \rangle q - \langle q, x_i \rangle p \rangle = \langle p, x_i \rangle \langle q, x_i \rangle - \langle q, x_i \rangle \langle p, x_i \rangle = 0.$$

Therefore,

$$\text{grad}_{x_i} \mathcal{L}_{\text{Sliced}}(X) = \nabla_{x_i} \mathcal{L}_{\text{Sliced}}(X).$$

## C  Convergence of the Lloyd Regularizer

The convergence of Lloyd with Riemannian SGD can be shown with a direct application of Bonnabel (2013, Theorem 1).

Let $V_X(i) := \{s \in \mathbb{S}_m : d(s, x_i) \leq d(s, x_j) \text{ for all } j \neq i\}$ denote the spherical Voronoi cell around cluster center $x_i$. The gradient $w.r.t.$ all parameters is:

$$\text{grad}_X \mathcal{L}_{\text{Lloyd}}(X) = \mathbb{E}_y[Q(X, y)] \tag{20}$$

or, blockwise $w.r.t.$ on centroid at a time,

$$\text{grad}_{x_i} \mathcal{L}_{\text{Lloyd}}(X) = \mathbb{E}_y[Q_i(X, y)] \tag{21}$$

where $Q_i(X, y)$ can be defined as:

$$Q_i(X, y) = \text{Log}_{x_i}(y) 1_{V_X(i)}.$$

The length of the Riemannian Log for the sphere is the geodesic distance, which is bounded:

$$\|Q_i(X, y)\| = d(x_i, y) 1_{V_X(i)} \leq \pi$$

and therefore for the total gradient we also have:

$$\|Q(X,y)\| \leq \pi\sqrt{n}.$$

As the sphere is compact and has injectivity radius $\pi$, and Q is bounded as shown, we are in the conditions of Theorem 1 of Bonnabel (2013), and therefore Riemannian SGD with step sizes satisfying $\sum_{t>0} \gamma_t^2 < \infty, \sum_{t>0} \gamma_t = \infty$ converges.

## D    Effect of Dimensionality

### D.1    Kernel sensitivity

Points on $\mathbb{S}^m$ concentrate near its equator, i.e., when $m$ grows, the angles between some $z_0$ and a uniform point on the sphere concentrate around 90 degrees. All kernels we consider appear to exhibit similar linear convergence in log-log space (see Figure 7), but some are more sensitive in this critical area. Spherical kernels are generally more sensitive than Euclidean ones, with the exception of RBF, which justifies using Euclidean RBF in high-dimensional cases, at least when the data starts close to a uniform distribution, despite more attractive properties of other kernels.

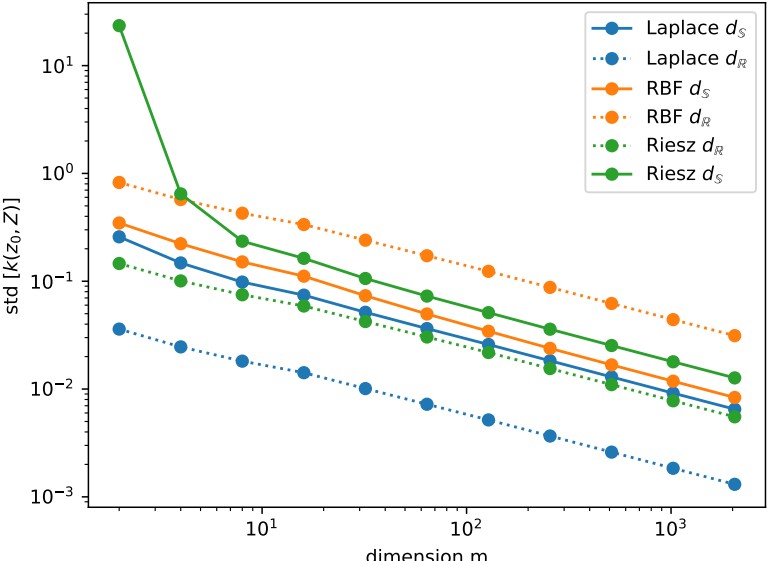

**Figure 7:**  Concentration of angles for different kernels.

### D.2    Additional results

To study the impact of dimensionality further we provide an additional empirical comparison of the minimum geodesic distance and spherical variance for $m \in (64, 256, 512)$ for $n = 10k$ and $n = 20$ in Figures 8 and 9. The maximum $d_{\min}$ increases while dimensionality grows. The red line shows the maximum achievable separation (90 degrees). In terms of spherical variance, all methods converge to 1.0 except for Lloyd.

## E    Additional experimental details and results

### E.1    Neural Machine Translation: Experimental Setup

For subword tokenization we used the same SentencePiece (Kudo & Richardson, 2018) model, specifically the one used in the MBart multilingual model (Liu et al., 2020). This choice allows for unified preprocessing for all

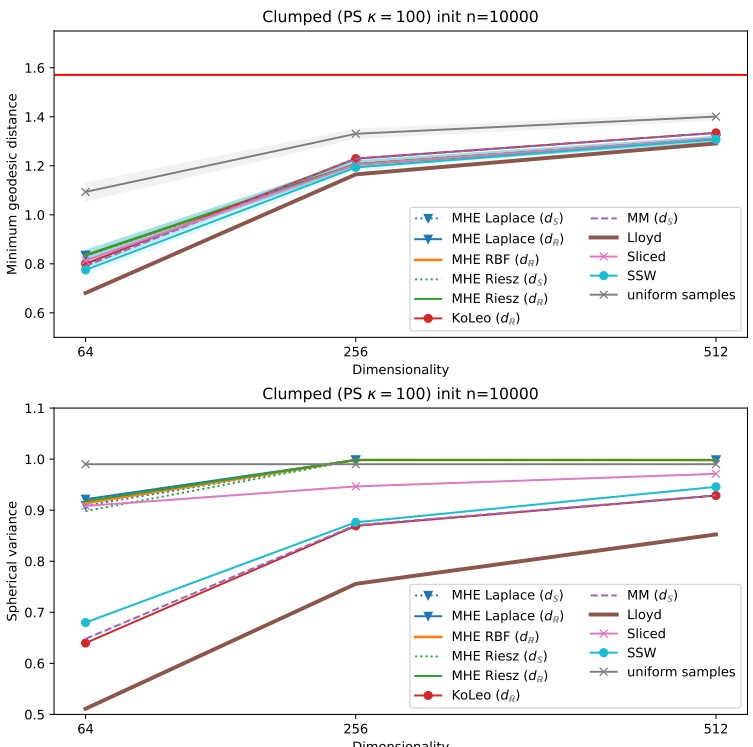

**Figure 8:** Maximum minimum angle and spherical variance for synthetic data with different dimensionalities and number of points 10000.

languages we cover. We used `fairseq` (Ott et al., 2019) framework for training our models. Baseline discrete models (Euclidean baseline) are trained with cross-entropy loss, label smoothing equal to 0.1 and effective batch size 65.5K tokens. All models are trained with learning rate $5 \cdot 10^{-4}$ and 10k warm-up steps for 50k steps in total. Spherical baseline and models with dispersion regularizer are trained by defining decoder's embeddings layer as a manifold parameter. We tune learning rate for Riemannian Adam (Becigneul & Ganea, 2019) in the range $[5 \cdot 10^{-5}, 5 \cdot 10^{-4}, 5 \cdot 10^{-3}]$ and report results with the learning rate $5 \cdot 10^{-3}$. We used SacreBLEU (Post, 2018) with the following signature `nrefs:1|case:mixed|eff:no|tok:13a|smooth:exp|version:2.3.1` and COMET (Rei et al., 2020) with `unbabel-comet` library version 2.2.2[5] and `Unbabel-wmt22-comet-da` model.

### E.2 Neural Machine Translation: Gradient Norms

We show in Figure 10 how gradient norms and minimum distances of target language embeddings vary throughout the training process. Note that at the step=0, the norms and minimum distances are the same.

### E.3 Riemannian vs Euclidean Optimization

We compare the results of optimal angle approximation using projected gradient Adam $\mathbb{R}^m$ against Riemannian Adam over $\mathbb{S}_m$. All other parameters are the same as described in §4.1. Results are shown in Figure 11. However, for all other methods except KoLeo we can see that Riemannian optimization performs better overall.

### E.4 Sample Convergence of the Sliced Regularizer

For better insight into the number of great-circle samples required to estimate $\mathcal{L}_{\text{Sliced}}$ in high dimensions, we perform a numerical experiment. Fixing $X$ to be a uniformly-sampled configuration of $n = 10k$ points

---

[5] https://github.com/Unbabel/COMET

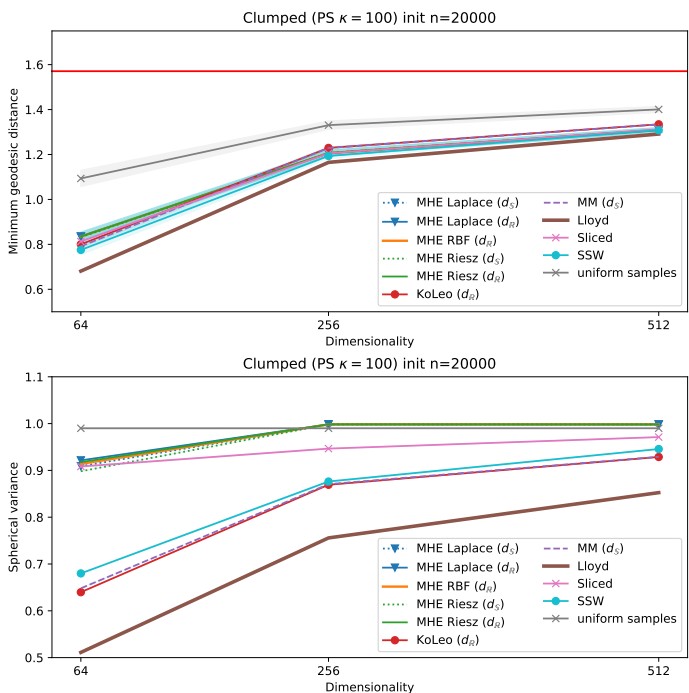

**Figure 9:** Maximum minimum angle and spherical variance for synthetic data with different dimensionalities and number of points 20000.

in dimension $m = 128$, we draw up to $10k$ uniformly-sampled great circles and report the Monte Carlo estimated average objective in Figure 12. The value converges relatively quickly and 1000 samples seem more than enough. In experiments, we find it sufficient to draw a single sample per update, because we perform sufficiently many updates.

### E.5 Spherical Sliced Wasserstein: Additional Results

We do a comparison between Sliced and SSW on a larger scale using synthetic data. We tune the number of great circles sampled in one iteration and the learning rate. We report the best results for different dimensionalities and the number of points in Figure 13. We can see that in lower dimensions and with a smaller number of points, SSW can achieve a better minimum geodesic distance than Sliced. But as number of points grows, they provide identical results.

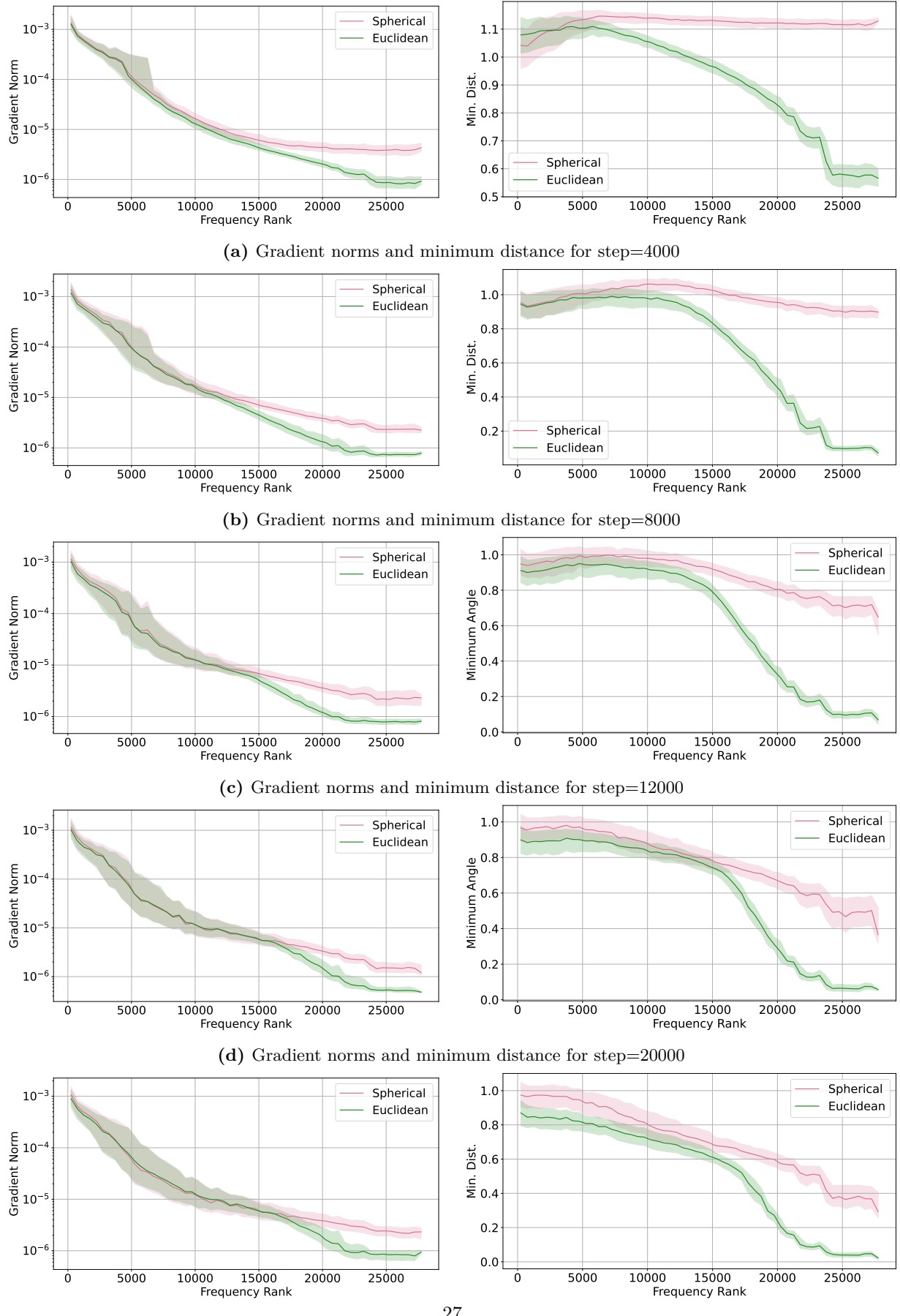

**(a)** Gradient norms and minimum distance for step=4000

**(b)** Gradient norms and minimum distance for step=8000

**(c)** Gradient norms and minimum distance for step=12000

**(d)** Gradient norms and minimum distance for step=20000

**(e)** Gradient norms and minimum distance for step=32000

**Figure 10:** Training dynamic of gradient norms and minimum distances of the target language embeddings.

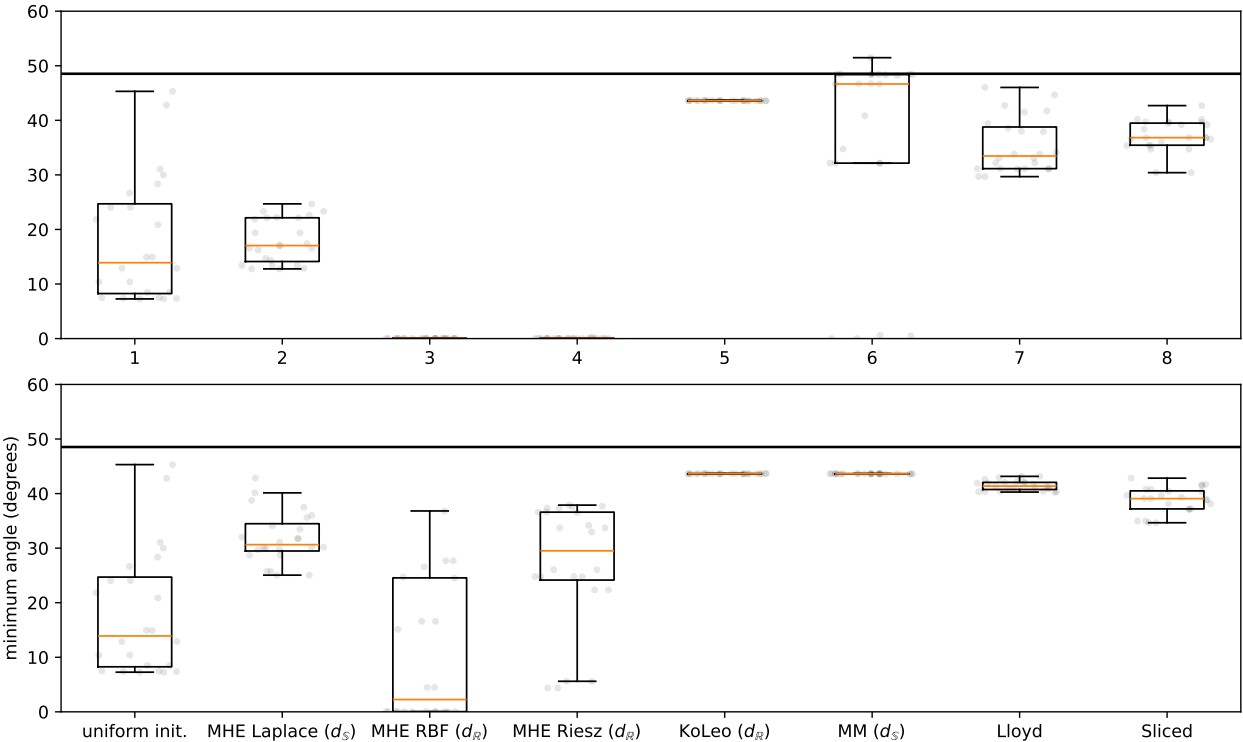

**Figure 11:** Minimum angles (degrees) for each of the N=24 points with respect to optimization methods. Top: projected gradient in Euclidean space. Bottom: Riemannian gradient on the sphere.

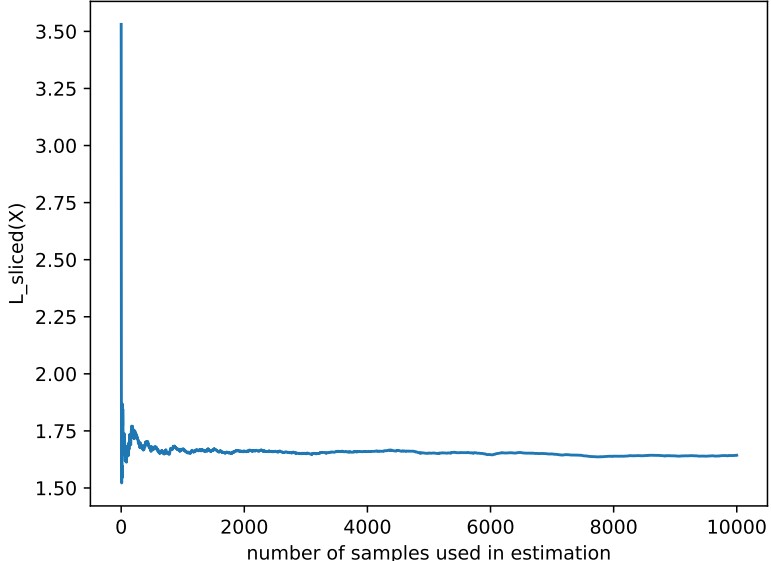

**Figure 12:** Convergence of the sliced regularizer.

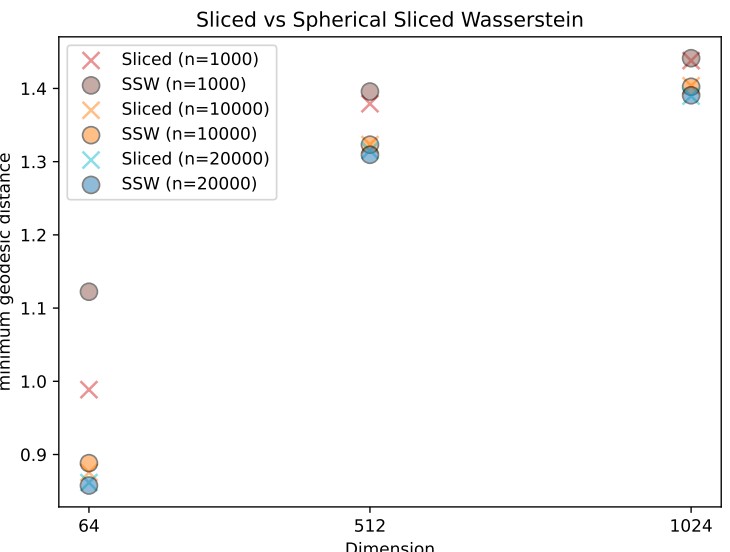

**Figure 13:** Minimum geodesic distance for different dimensions and number of points optimized by Sliced and SSW. Points generated with PS distribution with $\kappa$=100.

