# OpenReview forum: "Keep your distance: learning dispersed embeddings on $\mathbb{S}_{m}$"
_TMLR — Accepted by TMLR_

### Review · Reviewer_c9uQ · 2025-05-10

**Summary Of Contributions:**

This work studies the problem of learning dispersed embeddings on the sphere. The authors first provide an overview of different ways to measure the dispersion on the sphere, and of objectives that can be minimized to learn dispersed features. They make a connection between existing methods based on the minimization of an energy with kernels, and the minimization of the MMD on the sphere. They also make connections between these objectives and the Tammes problem. Additionally, relying on the observation that the optimal dispersion of $m$ points is know on a circle, they propose an objective based on the slicing process to enforce the dispersion in higher dimensional spaces. Finally, they compare the introduced methods on different applications, ranging from synthetic embeddings to image classification with prototypes, and neural machine translation.

**Audience:**

Yes

**Broader Impact Concerns:**

No.

**Claims And Evidence:**

Yes

**Requested Changes:**

In Section 2.3, and the description of criteria to measure dispersions, I believe that any distance between probability distributions on the sphere could be added (for instance in particular optimal transport distances such as the Wasserstein distance or the Spherical Sliced-Wasserstein distance [1]).

The spherical Sliced-Wasserstein distance introduced in [1] can be efficiently computed between any discrete distribution and the uniform measure on the sphere. Thus, it can be optimized to generate dispersed points (see e.g. https://pythonot.github.io/auto_examples/backends/plot_ssw_unif_torch.html#sphx-glr-auto-examples-backends-plot-ssw-unif-torch-py). I am wondering how it would compare with methods used in this paper?

The recent paper [2] compared several methods to sample uniform points on the sphere (e.g. quasi Monte Carlo methods...) to then approximate the integral with respect to the uniform measure on the sphere. I believe that any method of [2] could also be compared with the methods of this paper, and conversely, the methods of this paper could be used to estimate integral. I think a discussion (or even comparisons) about the methods from this paper would be a nice addition.

It could also be a nice addition to add a table somewhere summarizing some of the results, and with computational complexities.

In Table 5, there is a description of kernels on the sphere. But some of them are known to not be positive definite. It could be nice to add a column with this information. And a related question is, is it an issue?

In the experiments, why are you always sampling only 1 great circle in the sliced method? I guess it would not be much more costly to take several directions, and it should give a better approximation. Also, it would be nice to have a comparison between the proposed sampling method along axis, and the uniform one.

In Section 4.1, Figure 4 is not refered to.

In the related works Section, it is written that "DPPs are intended for selecting subsets of $k \le n$". It is true for finite DPPs, but there are also continuous DPPs, see e.g. [3].

Typos:
- Page 2: "the projection the standard"
- Equation (5): $\mathcal{L}\_{\mathrm{Max-Min}}$ should be $\mathcal{L}\_{\mathrm{MM}}$?


[1] Bonet, C., Berg, P., Courty, N., Septier, F., Drumetz, L., & Pham, M. T. (2022). Spherical sliced-wasserstein. arXiv preprint arXiv:2206.08780.

[2] Sisouk, K., Delon, J., & Tierny, J. (2025). A User Guide to Sampling Strategies for Sliced Optimal Transport. arXiv preprint arXiv:2502.02275.

[3] Fanuel, M., & Bardenet, R. (2021). Nonparametric estimation of continuous DPPs with kernel methods. Advances in Neural Information Processing Systems, 34, 24124-24136.

**Strengths And Weaknesses:**

This paper provides an interesting overview of methods which can be used to enforced dispersion on high dimensional spheres. It also provides nice connections between different objectives, such as energy and kernel methods, with the Tammes problem which is equivalent in the limit, and the MMD on the sphere. It also introduces a new sliced algorithm which seems to be very competitive with the other methods. Moreover, the comparison are done on several Machine Learning tasks. So overall, I believe this is a nice contribution to the community.


**Strengths:**
- Description of several criteria to measure dispersion, and several methods to enforce it
- Connection between different objectives
- New sliced algorithm demonstrating good results
- Comparison of the methods on several settings, interesting to the ML community.

**Weaknesses:**
- I believe that some methods which could also solve this kind of problem are lacking.
- This work lack some discussion about the complexity of the methods (or could be clearer), and if the different objectives come with optimization guarantees or not.

---

> ### Author Response · Authors · 2025-06-05
>
> Thank you for your review and providing us with related literature. We greatly appreciate that.
>
> > In Section 2.3, and the description of criteria to measure dispersions, I believe that any distance between probability distributions on the sphere could be added (for instance in particular optimal transport distances such as the Wasserstein distance or the Spherical Sliced-Wasserstein distance [1]). The spherical Sliced-Wasserstein distance introduced in [1] can be efficiently computed between any discrete distribution and the uniform measure on the sphere. Thus, it can be optimized to generate dispersed points (see e.g. https://pythonot.github.io/auto_examples/backends/plot_ssw_unif_torch.html#sphx-glr-auto-examples-backends-plot-ssw-unif-torch-py). I am wondering how it would compare with methods used in this paper?
>
> Indeed, we missed the part of [1] where they derive a closed-form expression for the uniform case, and that results in a viable sliced dispersion algorithm. Although it optimizes a different function, our preliminary experiments on synthetic data and the Tammes problem demonstrate that it performs similarly to our Sliced approach and offers additional useful theoretical insights. We include an empirical comparison to Appendix F, and we will include rigorous comparison with SSW in the next revision.
>
> [1] Bonet, C., Berg, P., Courty, N., Septier, F., Drumetz, L., & Pham, M. T. (2022). Spherical sliced-wasserstein. arXiv preprint arXiv:2206.08780.
>
> > The recent paper [2] compared several methods to sample uniform points on the sphere (e.g. quasi Monte Carlo methods...) to then approximate the integral with respect to the uniform measure on the sphere. I believe that any method of [2] could also be compared with the methods of this paper, and conversely, the methods of this paper could be used to estimate integral. I think a discussion (or even comparisons) about the methods from this paper would be a nice addition.
>
> Thank you for sharing this exciting work. Indeed, we have some preliminary results showing that sampling uniformly yields in some cases to a better minimum geodesic distance, especially in smaller dimensions (Please see Figure 10 and 11 in the updated manuscript). It has interesting implications, and we will include it in a discussion in the next revision.
>
> > It could also be a nice addition to add a table somewhere summarizing some of the results, and with computational complexities.
>
> That is a very good point. We have added Table 5 in Section 4.6, which includes the computational complexity for all regularizers.
>
> > In Table 5, there is a description of kernels on the sphere. But some of them are known to not be positive definite. It could be nice to add a column with this information. And a related question is, is it an issue?
>
> In the revised manuscript, we have added a column to Table 6 (former Table 5). Indeed, if the kernel is not PD, it affects theoretical results: as there is no induced RKHS, the MMD interpretation does not hold. However, empirically, this does not appear to pose problems, especially in large dimensions, where angles are more concentrated.
>
> > In the experiments, why are you always sampling only 1 great circle in the sliced method? I guess it would not be much more costly to take several directions, and it should give a better approximation. Also, it would be nice to have a comparison between the proposed sampling method along axis, and the uniform one.
>
> In Appendix A3, we provide more insights into the number of great circles we sample. To summarize, we find that with a large enough number of outer iterations, it is sufficient to sample a single slice per iteration.
>
> > In the related works Section, it is written that "DPPs are intended for selecting subsets of k≤n". It is true for finite DPPs, but there are also continuous DPPs, see e.g. [3].
>
> That is a good catch, thank you. We will revise the related work accordingly.
>
> We also thank you for noticing the typos and missing reference to the Figure. It is fixed in the new uploaded version.

---

### Review · Reviewer_Dr9k · 2025-05-18

**Summary Of Contributions:**

- The paper unifies a diverse family of hyperspherical‐dispersion objectives, including max–min distance, Kozachenko-Leonenko entropy, Minimum Hyperspherical Energy (MHE) and uniformity losses, by showing that they are all pair-wise distance functionals that differ only in the choice of potential or kernel, thereby clarifying their computational cost and optimization geometry.

- It proves that MHE with any positive-definite kernel is an unbiased estimator of the squared Maximum-Mean-Discrepancy (MMD) from the uniform measure on the sphere, giving a principled statistical interpretation to popular “energy” regularizers.

- Building on this insight, the authors introduce two new, computationally efficient dispersion regularisers: i) Online Lloyd regularizer, which treats dispersion as on-the-fly quantisation of $U(\mathbb{S}^m)$ and applies a stochastic Riemannian variant of Lloyd’s $k$-means algorithm within mini-batches; ii)  Sliced dispersion, which projects embeddings onto random great circles, computes the closed-form optimal 1-D code, and penalizes the orthogonal distance, yielding an $\mathcal{O}(n)$ per-batch method with an explicit stochastic Riemannian gradient.

- The work proposes manifold-aware optimization by demonstrating that Riemannian Adam updates on $\mathbb{S}^m$ provide smoother training dynamics and superior dispersion compared with Euclidean gradients followed by projection.

- Finally, the paper provides practical guidance on when to deploy each regularizer, noting that max–min excels in fine-tuning well-spread codes, while sliced dispersion is preferable for very large or initially clustered vocabularies, and online Lloyd offers a favorable speed-quality trade-off as a drop-in mini-batch loss.

**Audience:**

Yes

**Claims And Evidence:**

Yes

**Requested Changes:**

- Give formal convergence guarantees for both new objectives:  For online Lloyd, state assumptions on step–size, mini-batch sampling and curvature of $\mathbb{S}^m$; prove almost-sure convergence to a stationary point *or* provide an upper bound on the expected quantization error after $T$ updates. For sliced dispersion: show that the stochastic Riemannian gradient is unbiased and that, with a diminishing-step schedule, the expected squared gradient norm vanishes.
- Augment the MHE–MMD analysis with finite-sample variance control: Derive variance for a generic positive-definite kernel and give an asymptotic CLT. Use this to justify the regularization weight and the number of Monte-Carlo samples for the uniform-term estimate.
- Analyze high-dimensional behaviour:  Discuss how concentration of measure on $\mathbb{S}^m (m\to\infty)$  affects the repulsive force of commonly used kernels (Laplace, RBF). Provide bounds (or empirical plots) showing how the minimum achievable pairwise angle $\theta_{\min}$ scales with $m,n$ under each objective.
- Establish an approximation guarantee w.r.t. the Tammes optimum or explain its impossibility:  Either prove that each objective attains a constant-factor approximation in general $m,n$ or supply a hardness argument clarifying why only empirical proximity can be expected.

**Strengths And Weaknesses:**

## Strengths

- Theoretical unification and clarity: Shows that popular hyperspherical‐dispersion objectives (max–min, Kozachenko–Leonenko entropy, energy/MHE, “uniformity” loss) are all pair-wise potentials differing only by the kernel $k$. Provides a rigorous link between Minimum Hyperspherical Energy and MMD. This statistical interpretation unifies dispersion with kernel two-sample testing and offers principled hyper-parameter choices.
- Novel, computationally efficient objectives:  Online Lloyd regulariser introduces a mini-batch, Riemannian variant of spherical $k$-means that is linear in the mini-batch size. Sliced dispersion reduces computation from $O(n^2)$ to $O(n)$ per batch via random great-circle projections and has an analytic Riemannian gradient.
- Manifold-aware optimisation: Convincingly demonstrates that Riemannian Adam steps on $\mathbb{S}^m$  are smoother and avoid the oscillations seen with Euclidean gradients and projection.

## Weaknesses

- No convergence guarantees for the new regularisers: The online Lloyd objective is presented as a stochastic Riemannian variant of spherical $k$-means, yet the paper offers neither a proof of almost-sure convergence to a local optimum nor a bound on the expected quantisation error after $T$ updates. For sliced dispersion, the manuscript states that each step is a “stochastic Riemannian gradient”, but omits any analysis of step-size conditions under which the iterates remain stable or converge.
- Unbiasedness without variance control: The MHE–MMD theorem establishes that $\mathcal{L}_{\text{MHE},k}(X)$ is an unbiased estimator, but the authors do not derive finite-sample variance bounds or asymptotic normality, crucial for understanding regularization strength and hyper-parameter tuning.
- Dimension-dependent behavior is unanalyzed:  Theoretical claims are stated for any $m$, yet the manuscript lacks discussion of the *concentration of measure $\mathbb{S}^m$ when $m\to\infty$. In high dimensions, pairwise dot products concentrate around zero, which may decay the repulsive force of kernels such as Laplace or RBF; no bound quantifies this effect.
- Lack of optimality against Tammes code: While experiments show that max–min and Lloyd approach the optimal Tammes energy for $n\le 24$ on $\mathbb{S}^2$, the theory provides no guarantee that the proposed objectives achieve a constant-factor approximation to the Tammes optimum in general $m,n$.

---

> ### Author Response · Authors · 2025-06-05
>
> We appreciate the time and effort you have taken to help us improve the paper. Your review has been very insightful. We would like to address the requested revisions here.
>
> 1. Convergence Guarantees
>
> The convergence of Lloyd with Riemannian SGD can be shown with a direct application of [1, Theorem 1].
>
> Let $V_X(i) \coloneqq \{ s \in \mathbb{S}_m: d(s, x_i) \leq d(s, x_j)\ \text{for all}\ j \neq i\}$ denote the spherical Voronoi cell around cluster center $x_i$. The gradient w.r.t all parameters is:
>
> $$
> grad_{X} \mathcal{L}_{Lloyd}(X)=\mathbb{E}_y [Q(X, y)]
> $$
>
> or, blockwise wrt once centroid at a time,
>
> $$
> grad_{x_i} \mathcal{L}_{Lloyd}(X)
> = \mathbb{E}_y [Q_i(X, y) ]
> $$
>
> where we define
>
> $$
> Q_i(X,y) = Log_{x_i}(y) 1_{V_X(i)}.
> $$
>
> The length of the Riemannian Log for the sphere is the geodesic distance, which is bounded:
>
> $$
> \|Q_i(X,y)\| = d(x_i, y) 1_{V_X(i)} \leq \pi
> $$
>
> and therefore for the total gradient we also have:
>
> $$
> \|Q(X, y)\| \leq \pi \sqrt{n}.
> $$
>
> As the sphere is compact and has injectivity radius $\pi$, and Q is bounded as shown, we are in the conditions of Theorem 1 of [1], and therefore Riemannian SGD with step sizes satisfying $\sum_{t>0} \gamma_t^2 < \infty, \sum_{t>0} \gamma_t = \infty$ converges.
>
> For the sliced estimator, the gradient is unbiased as it is a standard Monte Carlo estimator of an expectation:
>
> $$
> \nabla \mathcal{L}\_{Sliced} (X) = \nabla \mathbb{E}\_{P,Q} [g(X, P, Q) ] \approx_{MC} \nabla \frac{1}{k} \sum_i g(X, p_i, q_i) = \frac{1}{k} \sum_i \nabla g(X, p_i, q_i)
> $$
> where
>
> $$
> g = (\theta_i|_{p_iq_i} - {\theta}^{\star}_i|\_{p_iq_i}) \frac{\langle{x_i}{p_i}\rangle {q_i} - \langle {x_i}{{q_i}}\rangle {p_i}}{\langle{x_i}{{q_i}}\rangle^2 + \langle{x_i}{{p_i}\rangle}^2}
> $$
>
> However, $\nabla g$ is unfortunately not bounded, and we cannot use the same standard result from [1]. This is due to the fact that the projection of x onto a great circle S_pq is not defined when x is orthogonal to both p and q. As far as we can tell, this problem is present and not addressed in the work of Bonet et al (2023), who use the same great circle projection, and yet both their and our works find empirical success, suggesting indeed room for a better analysis here. Thank you for pointing us toward this observation, we will explore it further.
>
> [1] Bonnabel, Silvére. “Stochastic Gradient Descent on Riemannian Manifolds.” *IEEE Transactions on Automatic Control* 58 (2011): 2217-2229.
>
> 2. High-dimensional Behavior
>
> We assume the concentration of measure phenomenon you refer to is the convergence of angles between pairs of random points on a sphere toward 90 degrees, with variance tightening as dimension increases. In Figure 9 (App C of revision) we estimate the variance of the kernels studied in this paper as a function of dimension. All kernels we consider appear to exhibit similar linear convergence in log-log space, but some are more sensitive in this critical area. Spherical kernels are generally more sensitive than Euclidean ones, with the exception of RBF, which justifies using Euclidean RBF in high-dimensional cases, at least when the data is close to uniform, despite more attractive properties of other kernels.
>
> We provide an additional empirical comparison of the minimum geodesic distance and spherical variance in high dimension in Figures 10 and 11 in Appendix C.
>
> 3. Approximation guarantee w.r.t. the Tammes
>
> > Establish an approximation guarantee w.r.t. the Tammes optimum or explain its impossibility: Either prove that each objective attains a constant-factor approximation in general m,n or supply a hardness argument clarifying why only empirical proximity can be expected.
>
> As far as we know, there are not many such approximation guarantees available, also in part due to the optimal objective value of the Tammes problem being hard to characterize. Theorem 4 in [6] provides a bound on the Tammes objective attained by optima of MHE, while Theorem 5 relates the optima of the two problems in the limit. The former seems promising, and we are exploring whether we can derive similar results for MM, Lloyd, or Sliced regularization, but we cannot provide such a result at the moment. We are not aware of hardness results in the literature either. We appreciate any additional literature pointers or suggestions in this direction.
>
> [6] Liu, W., Lin, R., Liu, Z., Xiong, L., Schölkopf, B., & Weller, A. (2021, March). Learning with hyperspherical uniformity. In International conference on artificial intelligence and statistics (pp. 1180-1188). PMLR.
>
> (Cont'd in the next comment)

---

> > ### Author Response · Authors · 2025-06-05
> > **Author response Part 2**
> >
> > > Augment the MHE–MMD analysis with finite-sample variance control: Derive variance for a generic positive-definite kernel and give an asymptotic CLT. Use this to justify the regularization weight and the number of Monte-Carlo samples for the uniform-term estimate.
> >
> > For this suggestion, we kindly request a few additional clarifications.
> >
> > For MHE-MMD, the terms involving the uniform measure (i.e., cross-terms p-u and the double expectation over u) are constant w.r.t. x (the c in Lemma 1), and we did not estimate this constant or seek closed-form expressions of it since we could not see applications for this. Could you clarify what you mean by “number of Monte-Carlo samples for the uniform-term estimate”?
> >
> > In terms of variance, this direction is promising. Since our MMD estimator in Lemma 1 treats the terms involving u as a constant (and thus does not need to estimate them), we can derive limits and bounds that are somewhat better than for the usual full estimator on MMD (involving samples from u as well). In particular, instead of Theorem 10 from [2], we get:
> >
> > $$
> > \mathrm{Pr}[ \mathrm{MMD}^2_u(\mathcal{F}, p, u) - \mathrm{MMD}^2(\mathcal{F}, p, u) > t] \leq \exp(-t^2 \lfloor n/2\rfloor / K^2)
> > $$
> >
> > i.e., the denominator inside the exp is smaller by a factor of 8 because our U-statistic is in terms of k itself, and thus bounded in [0,K]. In terms of variance, we can use the expressions of Serfling [3] as in [4] and [4] (as well as the CLT in [2] Corollary 16) for instance to obtain the asymptotic expression
> >
> > $$
> > \text{Var}[\widehat{MMD}_u(\mathcal{F}, p, u)] = \frac{4(n-2)}{n(n-1)}\left( E[\langle \varphi(x), \mu_X\rangle^2] - \|\mu_X\|^2\right) + O(n^{-2})
> > $$
> >
> > or the similar second-order expression. These expressions are much simpler than the full MMD case since only the $p$ terms are needed, and the terms involving $u$ are constants. However, we don’t quite see how to apply this to help in selecting any hyperparameters. If you would point us in the right direction we would be happy to explore this further.
> >
> > Note that for our applications (dispersion) $p$ is an empirical measure and thus (with full-batch optimization) our regularizer is exact, but these variance results could be used to analyze the variance of the minibatch estimator. We are working on a brief new section in the supplementary materials providing these results and references.
> >
> > [2] Gretton, A., Borgwardt, K. M., Rasch, M. J., Schölkopf, B., & Smola, A. (2012). A kernel two-sample test. *The Journal of Machine Learning Research*, *13*(1), 723-773.
> >
> > [3] Serfling, R. J. (1980). *Approximation Theorems of Mathematical Statistics*. Hoboken, New Jersey: Wiley.
> >
> > [4] Bounliphone, W., Belilovsky, E., Blaschko, M.B., Antonoglou, I., & Gretton, A. (2015). A Test of Relative Similarity For Model Selection in Generative Models. *CoRR, abs/1511.04581*.
> >
> > [5] Sutherland, D.J. (2019). Unbiased estimators for the variance of MMD estimators. *ArXiv, abs/1906.02104*.

---

### Review · Reviewer_18EA · 2025-05-23

**Summary Of Contributions:**

This paper studied the loss functions for learning embeddings on the Hypersphere. These loss functions work as regularisers in many tasks, e.g., machine translation, and help us to obtain embeddings that are evenly distributed on the hypersphere. The main contribution of this paper is that it experimentally evaluated the effectiveness of many regularizers on several tasks, such as machine translation and image classification.

**Audience:**

Yes

**Claims And Evidence:**

Yes

**Requested Changes:**

* Could you clarify the relationship between this paper and existing papers, and clarify which part the new ideas in this paper are?
* Can the authors add error bars in Tables 2-4?
* Although each regularizer has been compared experimentally, can the authors discuss the difference in the more intuitive advantages of each regularizer?

**Strengths And Weaknesses:**

## Strength
* This paper evaluated many regularizers in several settings, such as machine translation tasks and image classification tasks.

## Weakness
* The manuscript presents different methods across sections, especially in Sec. 3, and there is no discussion to compare them in Sec. 3. As a result, it becomes challenging to follow the overall argument.
* It was not clear how this paper relates to the previous study. This paper explains several regularizers in Sec. 3.1-3.3, but it is unclear whether these regularizers are existing or proposed methods. As far as I understood, the authors explained how we can use the existing loss, e.g., MMD, Eq. (8), as the regularizer for learning embeddings on the Hypersphere.
* At the beginning of Sec. 3, the authors claimed that "then define two novel dispersion objectives with more appealing computational properties", but I could not understand what the benefit is in terms of computational properties.
* In this paper, Tables 3 and 4 experimentally validate the effectiveness of many regularizations, but the improvement is very small, with only a gain of 1 or less in the BLUE score.
* For the experimental results shown in Sec. 4.4 and 4.5, the error bars were not shown. Since the improvement is small, it is important to evaluate these regularizers with several seed values and show whether these regularizers can consistently improve the scores.
* For the experimental results shown in Sec. 4.4 and 4.5, the stepsize was not tuned by grid search.

---

> ### Author Response · Authors · 2025-06-05
>
> Thank you so much for your review. We appreciate your suggestions and believe they will further strengthen the paper.
>
> > Could you clarify the relationship between this paper and existing papers, and clarify which part the new ideas in this paper are?
>
> 1. This paper reviews a range of dispersion objectives from various research areas and presents a unified perspective on them. To the best of our knowledge, most of the literature is disconnected, and connections are rarely (if at all) discussed.
> 2. We show a new connection between a known class of dispersion objectives and maximum mean discrepancy, justifying what this regularizer actually optimizes (not in a limit), and allowing us to transfer any known results about MMD to this estimator.
> 3. Moreover, we propose a new dispersion objective, the Lloyd regularizer, which has computational advantages compared to pairwise objectives in high-dimensional scenarios.
>
> > Can the authors add error bars in Tables 2-4?
>
> For MT and CoNMT we performed paired bootstrap resampling implemented in `sacrebleu` (https://github.com/mjpost/sacrebleu?tab=readme-ov-file#paired-significance-tests-for-multi-system-evaluation) and `comet-compare` (https://github.com/Unbabel/COMET?tab=readme-ov-file#comparing-multiple-systems). In the revised version, we explicitly show in Tables 3 and 4 which results are statistically significant over the baseline with a p-value<0.05.
>
> For image classification with prototypes, we are working on adding additional runs, and we will report the mean over three runs in the next revision.
>
> > Although each regularizer has been compared experimentally, can the authors discuss the difference in the more intuitive advantages of each regularizer?
>
> Pairwise objectives are a good choice when the full matrix of pairwise distances can be calculated efficiently, i.e., when the number of datapoints and/or dimensionality are relatively small. In practical ML applications, dimensionality and number of points are typically large, as in §4.4 and §4.5, and given the limitations of computational resources, we can either calculate pairwise distance for a random batch of datapoints or use alternatives like Lloyd and Sliced. We summarize this in Section 4.6 and give computational complexity in Table 5.
>
> > *For the experimental results shown in Sec. 4.4 and 4.5, the stepsize was not tuned by grid search.*
>
> We have tried a small range of the step sizes for the Riemannian optimizer [5e-5. 5e-4, 5e-3] and report the best results with a step size of 5e-3. We ensured that this was clarified in Appendix D.

---

### Author Response · Authors · 2025-06-05

We want to thank all reviewers for their feedback. We are very happy to receive such thoughtful suggestions. We have uploaded a new version of the manuscript and highlighted the new text with the red color for your convenience.

---

### Decision · Action_Editor_JGYc · 2025-07-02

**Recommendation:** Accept with minor revision

**Additional Comments:**

The paper was appreciated by the reviewers who has a few comments and questions. The response and discussion from the authors were very appreciated ans all reviewers agree that the paper should be published in TMLR. I agree and recommend acceptance but ask that the authors implement all the changes discussed in the responses and add the discussions to the paper for the camera ready revision.

More details:
- All the discussion with reviewer Dr9k about convergence, high dimension and SLT is particularly interesting and deserves to be part of the main paper.
- Answers to c9uQ were nice but not all the discussed changes are implemented in the current version.   Please include SSW discussion and results in the main paper instead of just in an Appendix since it is very relevant.

**Audience:**

Yes

**Audience Explanation:**

Dispersing data on the sphere is a classical and important part of modern representation learning and this paper is definitely of interest to the TMLR audience.

**Claims And Evidence:**

Yes

**Claims Explanation:**

The paper makes connection between different communities and propose a nice overview of strategies for dispersion on sphere with new contributions. Numerical experiments are rigorous and support the discussion in the paper.

---

> ### Author Response · Authors · 2025-07-30
>
> We are very grateful for the positive feedback on our work. Below, we provide a summary of the changes made in the camera-ready version of the paper:
> - Abstract&Introduction: Minor edits to accommodate the inclusion of SSW and to add a link to the code.
> - §2.3: Added a paragraph on Optimal Transport and Spherical Sliced Wasserstein (SSW).
> - §3.1: Included the definition of the constant in the MMD formulation, a discussion of variance properties with a reference to the Central Limit Theorem (Appendix A.3), and comments on high-dimensional behaviour with a pointer to the sensitivity analysis of kernels in high dimensions (Appendix D.1).
> - §3.3: Added a discussion on the convergence properties of Sliced and SSW regularizers.
> - §4.1, §4.2, §4.3: Included additional experiments for SSW.
> - §4.6: Added a discussion on the computational complexity of regularizers, along with a complexity comparison presented in Table 5.
> - §5: Included a discussion on the relationship between sampling and dispersion.
>
> We would like to once again thank all reviewers and the action editor for their valuable suggestions.